# SCALING SEQUENCE-TO-SEQUENCE GENERATIVE NEURAL RENDERING

**Shikun Liu**\*, **Kam Woh Ng**\*, **Wonbong Jang, Jiadong Guo, Junlin Han, Haozhe Liu,**
**Yiannis Douratsos, Juan C. Pérez, Zijian Zhou, Chi Phung, Tao Xiang & Juan-Manuel Pérez-Rúa**
Meta AI     \*Equal Contribution
{shikun,kamwohng}@meta.com

## ABSTRACT

We present Kaleido, a family of generative models designed for photorealistic, unified object- and scene-level neural rendering. Kaleido is driven by the principle of treating 3D as a specialised sub-domain of video, which we formulate purely as a sequence-to-sequence image synthesis task. Through a systemic study of scaling sequence-to-sequence generative neural rendering, we introduce key architectural innovations that enable our model to: i) perform generative view synthesis without explicit 3D representations; ii) generate any number of 6-DoF target views conditioned on any number of reference views via a masked autoregressive framework; and iii) seamlessly unify 3D and video modelling within a single decoder-only rectified flow transformer. Within this unified framework, Kaleido leverages large-scale video data for pre-training, which significantly improves spatial consistency and reduces reliance on scarce, camera-labelled 3D datasets — all without any architectural modifications. Kaleido sets a new state-of-the-art on a range of view synthesis benchmarks. Its zero-shot performance substantially outperforms other generative methods in few-view settings, and, for the first time, matches the quality of per-scene optimisation methods in many-view settings. For supplementary materials, including Kaleido's generated renderings and videos, please refer to our website: https://shikun.io/projects/kaleido.

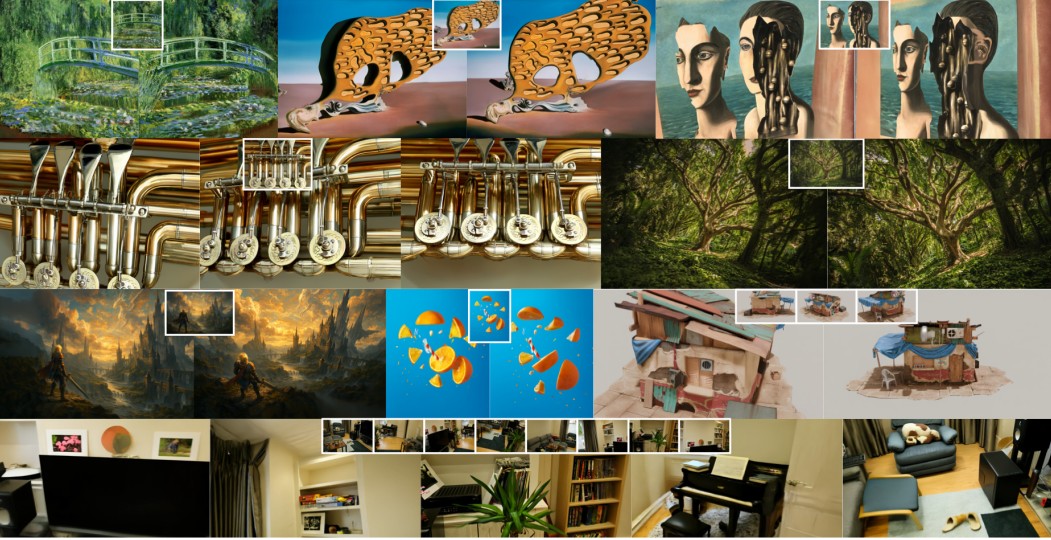

Figure 1: *Kaleido* is a generative rendering engine that creates photorealistic views in any style from any number of reference images (white boxes) with full 6-DoF camera control.

## 1 INTRODUCTION

Rendering and view synthesis are foundational to 3D computer vision and graphics, driving applications across virtual reality, cinematic effects, robotics, and autonomous driving. By allowing a scene

to be rendered from arbitrary viewpoints based on a limited set of reference views, view synthesis mimics the adaptability of human vision — the ability to construct and reconstruct a coherent 3D understanding of our surroundings.

While deep learning, fuelled by massive datasets and scalable model designs, has achieved great success in language and 2D vision, its progress in 3D vision for *general-purpose rendering* has been comparatively slow. We argue this stems from two persistent and interconnected bottlenecks:

1. *A Fragmented Landscape of 3D Representations.* 3D vision lacks a consensus on the *right 3D representation*, with methods spanning explicit structures like voxels (Wu et al., 2015), point-based representations such as point clouds (Qi et al., 2017; Guo et al., 2020) and 3D Gaussian Splatting (Kerbl et al., 2023), to implicit functions like neural fields (Mildenhall et al., 2020; Xie et al., 2022). This fragmentation has prevented the focused, collective effort required to scale a powerful architecture for any single representation, as development remains divided across incompatible data formats.

2. *The High Cost of 3D Data.* 3D datasets are scarce and difficult to obtain primarily because their creation is guided by the principle of *strict 3D consistency*. Achieving this level of precision requires either hand-crafting 3D synthetic meshes (Deitke et al., 2023b;a) or employing bundle adjustment and global alignments (Hartley & Zisserman, 2003) for slow multi-view camera labelling, making the data acquisition process slow, costly, and fundamentally difficult to scale.

As a direct consequence of these challenges, the 3D vision community has yet to converge on a scalable paradigm for 3D modelling. The combination of fragmented research efforts and restrictive data requirements has prevented the kind of focused, large-scale investment that enabled the dramatic architectural scaling and performance gains seen in language and 2D vision.

We believe these limitations, taken together, point to a fundamental oversight:

> *3D perception is not a geometric problem, but a form of visual common sense.*

The human ability to perceive 3D structure emerges from extensive observation of the world, not from maintaining a precise 3D model in the mind. For example, humans can interpret 3D geometry in optical illusions (*e.g.* in M.C. Escher's impossible structures and the Ponzo illusion), without having a physically accurate 3D representation or even a correct sense of depth. Accordingly, we argue that an ideal rendering system should not aim to *explicitly model perfect geometric consistency*, but instead to learn an *implicit representation* by capturing the statistical patterns of the extensive visual experience of the world.

Building on this insight, we introduce *Kaleido*, a scalable architecture for generative neural rendering. We design Kaleido as a type of *spatial generative model* that does not encode any explicit 3D structures. Instead, Kaleido inherits spatial perception and visual common sense directly from large-scale video data, purely in a data-driven way, similar to how modern large language models acquire textual common sense from large-scale corpora without relying on explicit linguistic rules. This leads to our central hypothesis, inspired by the success of domain-specific models fine-tuned from pre-trained language models (e.g., coding models in Roziere et al. (2023); Anil et al. (2023); Chen et al. (2021)), we believe that a powerful general-purpose rendering model can be created by treating *3D as a specialised sub-domain of video*. To put it simply,

> *We observed*      *large-scale corpus data → structured code data = a general-purpose coding model*
>
> ⇒   *We hypothesise*    *large-scale video data → structured 3D data = a general-purpose rendering model*

To realise this hypothesis, we reformulate rendering as a sequence-to-sequence problem, specifically as a pose-conditioned, image-to-image synthesis task. We first establish *a unified, geometrically consistent representation of space and time* as the core of our model design. This is achieved with a positional encoding design that extends the original Rotary Positional Encoding (RoPE) (Su et al., 2021) to parametrise all 2D, 3D, and temporal positions *relatively*, within the dot-product self-attention of a transformer (Vaswani et al., 2017). This foundational design enables Kaleido to learn rich world representations from large-scale, unstructured video data and then perform efficient transfer learning with much smaller-scale, structured multi-view 3D data, all within the same model *without any task-specific architectural changes*.

Building on this unified representation, Kaleido naturally benefits from scalable architectures and powerful generative techniques developed for language and vision. Specifically, Kaleido adopts a scalable transformer architecture inspired by Diffusion Transformer (DiT) (Peebles & Xie, 2023)

Figure 2: **Rendering as Sequence-to-Sequence Image Modelling.** We propose that neural rendering can be framed as a sequence-to-sequence task, unifying its design with language and video generation. In this formulation, a transformer (Vaswani et al., 2017) learns to generate image tokens conditioned on their spatial positions, similar to how language models condition on token positions in a sequence, and video models condition on temporal positions across frames.

and Llama-3 (Dubey et al., 2024), performing generative modelling via a rectified flow objective (Liu et al., 2023c; Esser et al., 2024) within a masked autoregressive framework (Li et al., 2024b; Fan et al., 2025; Liu et al., 2025a).

Finally, we identify that rectified flow SNR samplers commonly used for text-to-image/video generation are suboptimal for the precise pose conditioning required in rendering. We therefore introduce an improved, noise-biased sampling strategy and other key architectural adjustments to ensure stable and efficient scaling. Through extensive systemic studies, we validate these designs and highlight our primary contributions:

1. We introduce the *Kaleido family of Spatial Generative Models (SGMs)*, which can perform unified object- and scene-level view synthesis from any number of reference views to any number of target views with full 6-DoF camera control. This is enabled by the following designs:

   (a) A simple decoder-only rectified flow transformer that considers generative rendering as a sequence-to-sequence task.

   (b) A unified positional encoding design that seamlessly processes both 3D and video data within a single, unchanged architecture.

   (c) An effective scaling recipe for both model size and resolution, supported with a tailored SNR sampler and solutions for training instability.

2. Kaleido generates high-resolution images (up to 1024px) across diverse aspect ratios, achieving state-of-the-art results on numerous view synthesis and 3D reconstruction benchmarks. Most notably, in many-view settings, Kaleido is the first zero-shot generative model to match the rendering quality of per-scene optimisation methods such as Instant-NGP (Müller et al., 2022).

## 2 RELATED WORK

**Generative 3D Modelling and View Synthesis** Generative 3D modelling has rapidly evolved from synthesising isolated objects to composing entire, complex scenes. Pioneering text-to-3D works like Shap-E (Jun & Nichol, 2023) and Score Distillation Sampling (SDS) based methods like DreamFusion (Poole et al., 2022) laid the groundwork for single-object synthesis, inspiring a wave of research focused on high-fidelity object generation (Liang et al., 2024; Tang et al., 2024; Wang et al., 2023b; Shi et al., 2024). More recently, the frontier has expanded to scene generation, with approaches ranging from procedural construction (Sun et al., 2023; Raistrick et al., 2023) to direct compositional scene optimisation (Li et al., 2024a; Lu et al., 2024; Bai et al., 2023). A common thread in many of these works is the reliance on SDS to refine an explicit 3D representation.

Generative view synthesis models (Liu et al., 2023b;a;d; Shi et al., 2023) have emerged alongside this trend, but often face their own limitations. These methods typically struggle with multi-view consistency, are designed for a fixed number of reference (often one) and target views, and frequently rely on the same complex, two-stage SDS pipelines to enforce geometric coherence. Conversely, Kaleido's sequence-to-sequence design naturally handles an arbitrary number of both reference and target views, allowing it to generate spatially consistent views directly without requiring any post-processing or optimisation stages like SDS.

**Sequence-to-Sequence Generative View Synthesis**  Our work formulates generative view synthesis as a sequence-to-sequence modelling problem, built upon a pure transformer architecture. A critical challenge when applying transformers to this domain is effectively encoding camera positions. Recent advancements have introduced RoPE-style encodings (Su et al., 2021) to parameterise 6-DoF camera extrinsics, with notable examples including CaPE (Kong et al., 2024), GTA (Miyato et al., 2024), and also camera intrinsics in a more recent work (Li et al., 2025). Kaleido builds directly on this direction, leveraging a GTA-based framework to create a unified representation for both multi-view 3D poses and temporal video positions.

While other sequence-to-sequence methods like CAT3D (Gao et al., 2025), EscherNet (Kong et al., 2024) and SEVA (Zhou et al., 2025) have shown impressive results, their foundations lie in text-to-image latent diffusion models that use U-Net backbones. This reliance on a convolutional architecture is known to scale less effectively than pure transformers. Furthermore, these models often require 3D-specific learnable components, such as Plücker ray encodings for camera poses and a separate vision encoder for reference views. In contrast, Kaleido adheres to a pure transformer design from first principles, which results in a simpler, cleaner design that unifies 3D and video modelling, without any 3D-specific architectural modifications.

**Camera-Conditioned Video Generation**  Kaleido relies on video data for large-scale pre-training and is therefore conceptually related to a recent line of work on camera-conditioned video generation. However, the two fields address fundamentally different tasks. Camera-conditioned video generation is designed to tackle the conditional temporal prediction problem, typically on continuous camera trajectories and often with strong constraints. For instance, MotionCtrl (Wang et al., 2024b) and CameraCtrl (He et al., 2025a) are text-to-video pipelines that do not use reference images as NVS models do. CameraCtrl II (He et al., 2025b) can perform image-to-video, but it is constrained to a single reference image, which must also be the starting frame. ReCamMaster (Bai et al., 2025) requires the reference and target sequences to be of the same temporal length.

In contrast, Kaleido is a generative neural rendering model that handles a *3D spatial problem*. Its goal is to perform flexible "any-to-any" view synthesis: generating any number of target views from arbitrary 6-DoF poses, conditioned on any number of reference views. Kaleido's task is therefore not directly comparable to these 4D (3D+time) temporal prediction models. We do believe, however, that a unified model which can flexibly control both space and time is an important and exciting direction for future work.

Additional related works are listed in Appendix A.

## 3 KALEIDO: THE DESIGN DETAILS

### 3.1 BACKGROUND AND NOTATIONS

Kaleido considers rendering and video generation within a unified sequence-to-sequence framework, which is to estimate the conditional distribution of a set of target views given a set of reference views:

$$\mathcal{X}^T \sim p(\mathcal{X}^T | \mathcal{X}^R, \mathcal{P}^R, \mathcal{P}^T) \tag{1}$$

Here, the conditioning set consists of $N$ reference views $\mathcal{X}^R = \{x_{i=1:N}^R\}$ and their corresponding positions $\mathcal{P}^R = \{P_{i=1:N}^R\}$. The target set consists of $M$ target views $\mathcal{X}^T = \{x_{j=1:M}^T\}$ and their positions $\mathcal{P}^T = \{P_{j=1:M}^T\}$. The positions $P$ are defined flexibly depending on their data modality. For 3D data, each $P \in SE(3)$ represents a 6-DoF camera pose. For video data, each $P \in \mathbb{N}$ represents a temporal position (*i.e.*, a frame index). For efficiency, within each iteration, we sample a fixed total of $V$ views, and choose $N$ reference and $M$ target views such that $N + M = V$.

Kaleido is a latent rectified flow model (Rombach et al., 2022; Ma et al., 2024; Esser et al., 2024) that operates on spatially compressed image tokens. We first use a pre-trained VAE (Kingma & Welling, 2014) (with an $8 \times 8$ compression rate and 16 latent channels) to encode all reference and target images into a latent space: $\{\mathcal{Z}^R, \mathcal{Z}^T\} = \mathcal{E}(\{\mathcal{X}^R, \mathcal{X}^T\})$.

Following the rectified flow formulation (Liu et al., 2023c; Lipman et al., 2023), we then construct a linear interpolation path between each target latent $z^T \in \mathcal{Z}^T$ (from the data distribution $p_0$) and a standard normal noise latent $\epsilon \sim \mathcal{N}(0, I)$ (from the noise distribution $p_1$):

$$\mathcal{Z}_t^T = (1-t)z^T + t\epsilon, \quad \text{where } t \in [0, 1] \text{ and } \forall z^T \in \mathcal{Z}^T. \tag{2}$$

| | Objaverse (PSNR) | | uCO3D (PSNR) | | Throughput (# batches / sec.) |
|---|---|---|---|---|---|
| | 1 Ref. -> 5 Tar. | 5 Ref. -> 5 Tar. | 1 Ref. -> 5 Tar. | 5 Ref. -> 5 Tar. | |
| Baseline | 12.17 | 20.02 | 14.66 | 16.77 | 160 |
| (i) Architecture Design
*DiT -> Llama 3 (SwiGLU + GQA)* | 13.02 | 21.23 | 14.63 | 17.27 | 160 |
| (ii) Spatial Positional Encoding
*2D RoPE + 3D CaPE -> GTA [2D RoPE + 3D CaPE]* | 11.93 | 22.03 | 13.65 | 17.54 | 148 |
| (iii) View Sampling Strategy
*Fixed 6 to 6 -> Exp. Sampling w/ Masking* | 15.13 | 21.11 | 15.16 | 17.83 | 148 |
| (iv) Temporal Attention Design
*Temp. Attention [K=1] -> Temp. Win. Atten. [K=4]* | 15.73 | 21.79 | 15.55 | 18.45 | 142 |
| (v) Auxiliary Features
*None -> DINOv2 [DiT-B]* | 15.86 | 21.90 | 15.81 | 18.81 | 138 |
| (vi) Timestep Condition Design
*AdaLN-Zero [Top 1 Act.: 15192]* | 15.86 | 21.90 | 15.81 | 18.81 | 138 |
| (vii) Attention Registers
*No Registers [Top 1 Act.: 15192] -> 1 Register [Top 1 Act.: 397.7]* | 15.93 | 22.12 | 15.77 | 19.08 | 138 |
| (viii) Timestep Training Sampling
*LogitNormal [0, 1] -> Mode [Scale=0.8, Shift=3]* | 18.19 | 23.75 | 16.03 | 19.11 | 138 |
| (ix) Timestep Inference Sampling
*Linspace -> LinearQuadratic* | 18.09 | 23.95 | 17.03 | 19.79 | 138 |
| (x) with Video Pre-training
*No Pre-training -> Pre-training 200K Steps (2x Eff.)* | 18.28 | 24.60 | 17.18 | 20.15 | 138 |

Figure 3: **Kaleido Design Ablations.** We report PSNR and training throughput and evaluate performance in two settings: 5 to 1 and 5 to 5 reference to target views. We broadly split our designs into four categories: the Kaleido architecture design spaces (i–v); scaling stability techniques to handle large activations (vi–vii); training and inference timestep sampling strategies (viii–ix); and the role of video pre-training (x). The arrow ($\rightarrow$) indicates the progression from our initial baseline design to our final, optimised design choice.

A vision transformer (Dosovitskiy et al., 2020) then processes the combined sequence of clean reference latents $\mathcal{Z}^R$ and noised target latents $\mathcal{Z}_t^T$. We tokenise the latents using a patch size of $2 \times 2$ (for a combined spatial compression of $16 \times 16$), which we found provides an optimal trade-off between generation quality and inference speed. Kaleido is trained with a standard flow matching objective (Lipman et al., 2023), applied only to the target latents $\mathcal{Z}_t^T$, to estimate a velocity field between $p_0$ and $p_1$, conditioned as defined in Eq. 1, as a form of masked autoregression (Li et al., 2024b). To analyse the scalability, we present three model variants: **Kaleido-Small**, **Kaleido-Medium**, and **Kaleido-Large (Kaleido)**, with detailed architectures and training strategies introduced next.

## 3.2 KALEIDO ARCHITECTURE DETAILS AND TRAINING STRATEGIES

In this section, we present a comprehensive ablation study to identify the key architectural and training strategies for scaling sequence-to-sequence generative neural rendering. To facilitate this exploration efficiently, we used our Kaleido-Small model for all experiments, training each configuration for 100K steps on a mixture of the synthetic **Objaverse** (Deitke et al., 2023b) and the in-the-wild **uCO3D** (Liu et al., 2025b) datasets. Our main findings from this greedy search are summarised in Fig. 3, with the final architecture shown in Fig. 4. Full quantitative results and explorations of alternative designs are detailed in Appendix D, with additional design details in Appendix B.

### 3.2.1 DESIGNING KALEDIO'S DESIGN SPACES

We begin by exploring Kaleido's architectural design spaces and training strategies. Our Kaleido-Small's starting point is a vanilla DiT-L/SiT-L architecture (Peebles & Xie, 2023; Ma et al., 2024) within a rectified flow framework, whose scaling properties have been well established in image and video generation (Esser et al., 2024; Polyak et al., 2024; Chen et al., 2025).

**(i) Improved Architecture Design with Llama 3** We first incorporate recent architectural advances from state-of-the-art sequence-to-sequence language models like Llama-3 (Dubey et al., 2024). Specifically, we replace the standard GLU activations in our transformer's feed-forward layers with **SwiGLU** (Shazeer, 2020) and swap multi-head attention (MHA) for the more efficient **grouped-query attention (GQA)** (Ainslie et al., 2023). These simple modifications yield consistent performance gains across our experiments without increasing computational overhead.

**(ii) Unified Positional Encoding for Space and Time** One of the critical design decisions in Kaleido is a unified positional encoding that seamlessly represents 2D, 3D, and temporal positions within a single, consistent design. Specifically, we introduce a parameter-free encoding scheme that extends the principles of **RoPE-style relative encodings** (Su et al., 2021) and **Geometric Transfor-**

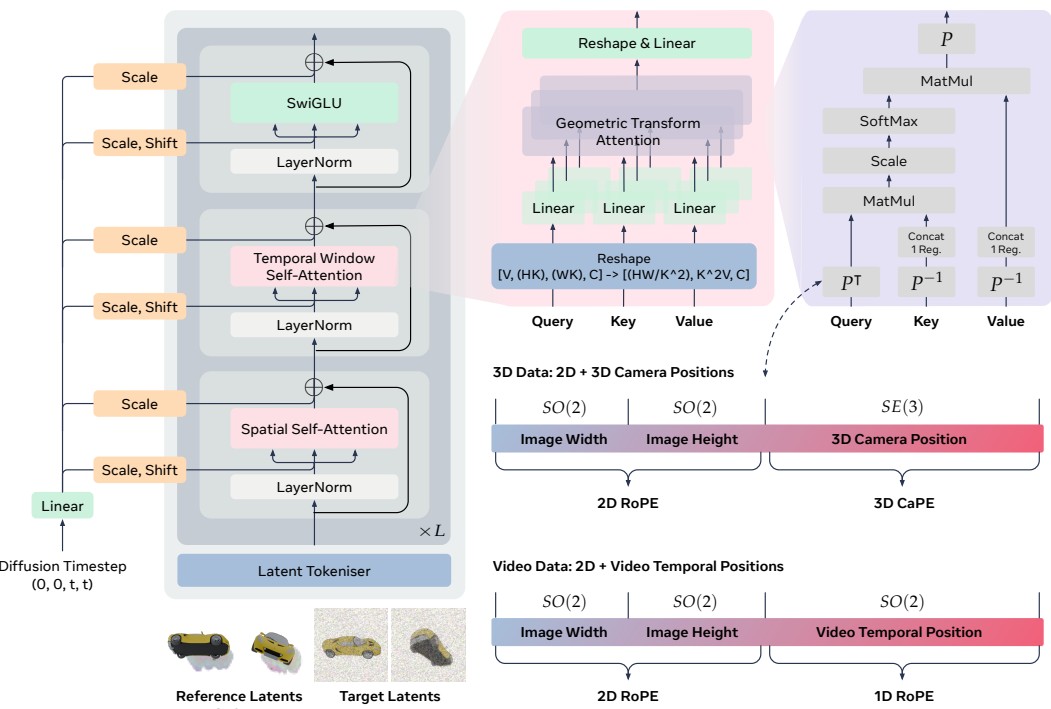

Figure 4: **Kaleido Architecture Design Details.** Kaleido is a simple, scalable decoder-only transformer that processes clean reference and noised target latents for sequence-to-sequence generative neural rendering. During training, a single timestep $t$ is sampled per scene and integrated into the network via AdaIN layers. Within each attention block, we apply a unified positional encoding, which consistently represents all 2D, 3D, and temporal positions. This enables a single architecture to seamlessly learn from both video and multi-view 3D data without modification.

mation Attention (GTA) (Miyato et al., 2024), which we adapt and generalise to create a unified representation for space and time, with all positional transformations constructed within a *bounded* range. This design allows Kaleido to process both multi-view 3D and video data without any architectural modifications. Our ablations confirm that this unified design outperforms (more significantly in multi-view settings) both a simpler baseline (2D RoPE + 3D CaPE without value-transformation) and the Plücker embeddings used in other leading sequence-to-sequence rendering models (Gao et al., 2025; Zhou et al., 2025; Jin et al., 2024). Detailed designs are illustrated in Appendix B.1.

**(iii) Principled View Sampling Improves Generalisation**   To improve generalisation, we introduce a new view sampling strategy, an *often-overlooked design aspect* in prior models that typically use fixed-view training (Jin et al., 2024; Kong et al., 2024; Gao et al., 2025). Our key insight is that *the rendering task is more constrained and easier with more reference views*; therefore, training should emphasise the *more challenging few-view scenarios*. To achieve this, we propose an **exponential sampling distribution**, $\pi(n) = \lambda e^{-\lambda n}$ where $\lambda = \ln(2)$, where its probability density halves as the number of reference views $n$ increases. By combining this distribution with random attention masking, our model is exposed to all possible combinations of $(n, m)$ reference and target view pairs such that $n + m \in [2, V]$. Our experiments confirm that this approach significantly outperforms both fixed-view and uniform sampling, providing a superior trade-off between single- and multi-view conditioning.

**(iv) Expanded Perception Field with Window Attention**   Our baseline model processes a token sequence of shape $V \times H \times W$ using a factorised attention mechanism: Spatial Attention followed by Temporal Attention, which efficiently processes token sequences but limits cross-view interactions. To improve this, we introduce **windowed cross-view attention**, where each query token attends to a local $K \times K$ window in all other frames, significantly enhancing cross-view feature exchange. This design maintains scalability, with a complexity of $O(V^2 K^4)$ that is far more efficient than full attention $O(V^2 H^2 W^2)$. Our experiments show consistent performance gains with larger window sizes, leading us to use $K = 4$ for our Small/Medium models and $K = 8$ for our Large model.

**(v) Integration of Auxiliary Visual Features** We study the integration of auxiliary visual features from pre-trained networks to enhance 3D perception. Our findings show that features from **DINOv2** (Oquab et al., 2024) further improve Kaleido's depth estimation on in-the-wild images, leading to more accurate renderings. These pre-trained semantic features performed similarly to, and sometimes slightly better than, pre-computed depth or surface normals, which encode explicit scene geometry built on top of the same DINOv2 model. We also observed that larger DINOv2 models provide additional, albeit marginal, performance gains. Based on this trade-off, we pair the feature extractor with our model size: we use DINOv2 with ViT-B backbone for Kaleido-Small and Medium, and DINOv2 with ViT-L backbone for our Large model.

### 3.2.2 MASSIVE ACTIVATIONS IN RECTIFIED FLOW TRANSFORMERS

During our initial scaling experiments, we traced severe training instability on high-resolution images to the emergence of massive activations within the transformer layers. While massive activations have been studied extensively in language models (Sun et al., 2024) (often called "attention sinks" (Xiao et al., 2024; Gu et al., 2025)) and in visual representation learning (Darcet et al., 2024), their behaviour within diffusion or rectified flow models remains largely unexplored. In this section, we provide the first empirical findings in this context, illustrated in Fig. 5.

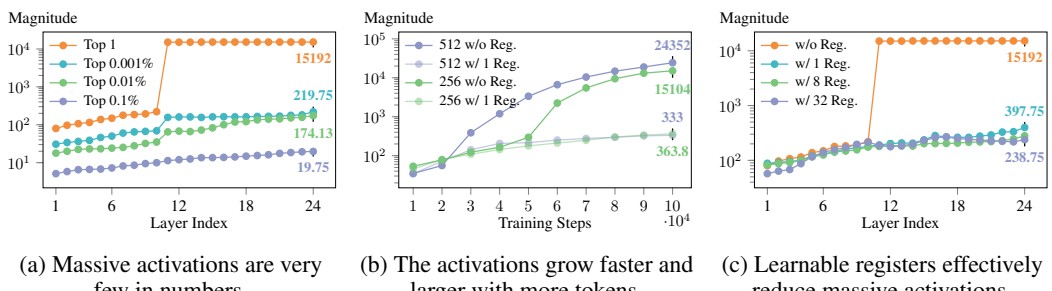

(a) Massive activations are very few in numbers.

(b) The activations grow faster and larger with more tokens.

(c) Learnable registers effectively reduce massive activations.

Figure 5: **Visual Analysis of Massive Activations in a Rectified Flow Transformer.** We analyse the massive activation phenomenon in our model. (a) We show that massive activations are sparse. (b) They grow aggressively with model depth and training time. (c) Appending learnable register tokens effectively suppresses these activations to a stable, low magnitude.

We also found that these massive activations emerge most prominently when training on a mixture of synthetic and real-world data, where they act as *global information aggregators* as observed in ViTs (Darcet et al., 2024) to reconcile the different rendering logic required for each domain. To resolve the training instability, we adopt the solution from Sun et al. (2024) and append **a single, learnable "register" token** to the keys and values in each attention layer, as confirmed in **ablation (vii)**. This simple modification proved highly effective, consistently reducing activation magnitudes to a stable level of ∼300. While we explored alternative solutions in **ablation (vi)**, such as modifying the timestep conditioning (Bozic et al., 2021; Sun et al., 2025; Tang et al., 2025), they were far less effective and resulted in lower overall performance.

### 3.2.3 TAILORED RECTIFIED FLOW SNR SAMPLERS FOR GENERATIVE RENDERING

Prior rectified flow models for text-to-image/video generation (Esser et al., 2024; Polyak et al., 2024) typically use logit-normal sampling (Atchison & Shen, 1980) to focus training on intermediate timesteps $t \in [0, 1]$. However, we found this approach to be suboptimal for neural rendering. We hypothesise this is because rendering is a *highly constrained generative problem*, where the output must be consistent with reference images, unlike the *much less constrained* nature of text-to-image/video synthesis. This insight suggests that rendering models should focus more heavily on the **early, high-noise timesteps** where the coarse scene structure is established.

Our ablations validate this hypothesis. In **ablation (viii)**, we found that sampling distributions shifted towards the noise end consistently outperform the standard baselines by a large margin. While a heavily shifted uniform distribution performed well, we choose **shifted Mode sampling** as our final training sampler design because it provides a superior balance between the critical early timesteps and the intermediate steps. To complement this, in **ablation (ix)**, we align our inference process by adopting a **linear-quadratic schedule**, which concentrates more denoising steps in the initial, high-noise part of the trajectory. We found that this design combination, using a noise-biased

sampler for both training and inference, delivered the single most significant performance improvement across all of our Kaleido design ablations. Detailed designs are illustrated in Appendix B.2.

### 3.2.4 VIDEO PRE-TRAINING IMPROVES 3D EFFICIENCY

Finally, in **ablation (x)**, we validate our core hypothesis of treating 3D as a sub-domain of video. The results confirm that pre-training on large-scale video data significantly boosts the efficiency of 3D fine-tuning, with 100K and 200K pre-training steps yielding 1.3x and 2.0x improvements, respectively. This demonstrates that a stronger video foundation model directly translates to faster convergence on view synthesis tasks.

### 3.3 FRAME INTERPOLATION AS ZERO-SHOT SPATIAL UPSAMPLER

Since multi-view 3D data lacks the temporal consistency required for typical video VAEs, Kaleido uses an image-based VAE, which makes generating dense video sequences computationally expensive. To overcome this, we employ an efficient two-stage pipeline: Kaleido first renders a sparse set of keyframes along a camera path, and then a separate, lightweight interpolation model based on FiLM (Reda et al., 2022) predicts the intermediate frames. This hybrid approach mitigates high memory costs and effectively emulates a temporal video decoder, allowing for the efficient production of smooth, high-frame-rate video sequences.

## 4 EXPERIMENTS

We designed three variations of our model: Small, Medium, and Large, with increasing parameter counts to demonstrate the scalability of our architecture. The design choices for each model are summarised in Table 1. Hereafter, we refer to our largest model simply as *Kaleido* and the entire collection as the *Kaleido family*.

Table 1: **Kaleido Family Architecture Details.** We detail the key hyper-parameters: the number of layers, hidden embedding size, number of query and key/value heads, the window size used in temporal attention, the choice of auxiliary DINOv2 encoder, and the total parameter count.

|                | Layers | Hidden Size | Query Heads | KV Heads | Window Size | Aux. Encoder    | Total Params. |
|----------------|--------|-------------|-------------|----------|-------------|-----------------|---------------|
| **Kaleido-Small**  | 24     | 1024        | 16          | 4        | 4           | DINOv2-B (86M)  | 653M          |
| **Kaleido-Medium** | 32     | 1280        | 20          | 5        | 4           | DINOv2-B (86M)  | 1.2B          |
| **Kaleido**        | 40     | 1792        | 28          | 7        | 8           | DINOv2-L (300M) | 3.1B          |

The Kaleido family is trained on a diverse mixture of object-level and scene-level datasets, featuring synthetic 3D object mesh renderings and real-world indoor and outdoor scenes. We outline our detailed training configurations and evaluation strategies in Appendix E.

### 4.1 RESULTS ON NOVEL VIEW SYNTHESIS

**Compared to Generative NVS Methods** We first evaluate Kaleido's zero-shot performance on standard novel view synthesis (NVS) benchmarks. For object-level synthesis, we compare against leading methods such as **SV3D** (Voleti et al., 2024) and **EscherNet** (Kong et al., 2024) on datasets including **OO3D** (Wu et al., 2023), **GSO-30** (Downs et al., 2022), and **RTMV** (Tremblay et al., 2022). For scene-level synthesis, we compare against the state-of-the-art rendering model **SEVA** (Zhou et al., 2025) on **LLFF** (Mildenhall et al., 2019), **Mip-NeRF 360** (Barron et al., 2022), and **Tanks and Temples** (Knapitsch et al., 2017). To ensure fair comparisons, we match evaluation resolutions and follow SEVA's strategy for handling scale ambiguity in single-view renderings.

In Table 2, our results demonstrate that Kaleido is both highly efficient and scalable. Even our smallest model, Kaleido-Small (0.6B), matches or surpasses larger baselines like SEVA (1.5B) and SV3D (2.3B). As we scale up, performance consistently improves, with our largest model decisively outperforming all competitors, achieving gains as high as +7.8 dB PSNR on GSO-30 (10 views) and +1.3 dB PSNR on Mip-NeRF 360 (6 views). Full SSIM/LPIPS metrics are provided in the Appendix F, and with additional high-resolution generations are shown in Fig. 6.

**Compared to Per-Scene Optimisation Methods** Next, we evaluate the upper bound of Kaleido's rendering precision when provided with many reference views. For this analysis, we compare against two state-of-the-art scene-specific optimisation methods: **Instant-NGP** (Müller et al., 2022) and **3D Gaussian Splatting (3DGS)** (Kerbl et al., 2023). As these methods are optimised per-scene, they represent a strong performance ceiling. We also include our generative NVS baselines that can

Table 2: **Zero-shot PSNR Performance with Generative Methods.** Kaleido achieves state-of-the-art NVS performance across all object- and scene-level benchmarks, with particularly dominant results in many-view settings. Notably, our Kaleido-Small model consistently matches or outperforms all baselines despite having significantly fewer model parameters.

| | OO3D | GSO-30 | | | | | RTMV | | | | | LLFF | | Mip-NeRF 360 | | | Tanks and Temples | | | |
|---|---|---|---|---|---|---|---|---|---|---|---|---|---|---|---|---|---|---|---|---|
| # Ref. Views | 1 | 1 | 2 | 3 | 5 | 10 | 1 | 2 | 3 | 5 | 10 | 1 | 3 | 1 | 3 | 6 | 1 | 3 | 6 | 9 |
| Eval. Data Type | Object | Object | | | | | Multi-Object | | | | | Scene | | Scene | | | Scene | | | |
| Eval. Resolution | 512 | 256 | | | | | 256 | | | | | 512 | | 512 | | | 512 | | | |
| Eval. Tar. Views | 20 | 15 | | | | | 10 | | | | | 5 | | 27 | | | 35 | | | |
| SoTA Model | SV3D | EscherNet | | | | | EscherNet | | | | | SEVA | | SEVA | | | SEVA | | | |
| Results (PSNR↑) | 19.28 | 20.24 | 22.91 | 24.09 | 25.09 | 25.90 | 10.56 | 12.66 | 13.59 | 14.52 | 15.55 | 14.03 | 19.48 | 12.93 | 15.78 | 16.70 | 11.28 | 12.65 | 13.80 | 14.72 |
| **Kaleido-Small** | 19.77 | 18.58 | 23.73 | 26.20 | 29.11 | 31.66 | 13.57 | 17.18 | 18.41 | 19.97 | 21.75 | 14.57 | 19.30 | 12.75 | 15.81 | 17.07 | 11.40 | 13.13 | 14.13 | 15.20 |
| **Kaleido-Medium** | 20.78 | 20.32 | 25.78 | 28.01 | 30.74 | 32.94 | 13.78 | 18.07 | 19.41 | 21.09 | 22.73 | 14.86 | 20.40 | 14.17 | 16.47 | 17.80 | 11.36 | 13.04 | 14.43 | 15.47 |
| **Kaleido** | 21.06 | 20.94 | 26.31 | 28.89 | 31.37 | 33.74 | 14.66 | 18.48 | 19.69 | 21.13 | 23.04 | 15.34 | 20.71 | 13.74 | 16.78 | 18.03 | 11.79 | 13.20 | 14.61 | 15.88 |

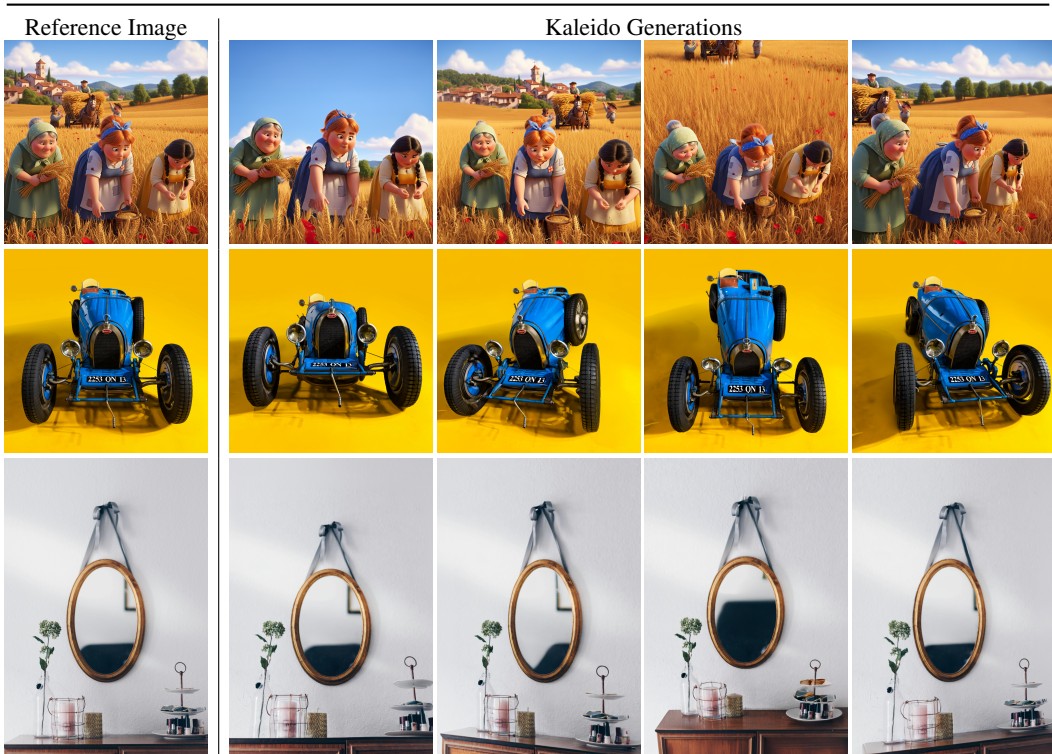

Figure 6: **In-the-Wild Single-View Rendering with Kaleido.** We showcase Kaleido's zero-shot generative capabilities on challenging in-the-wild images. From a single input view (first column), Kaleido generates a sequence of photorealistic novel views along a circular, object-centric camera trajectory. The examples feature complex scenes with diverse objects and structures, demonstrating Kaleido's remarkable generalisation and high-fidelity rendering quality.

accept a flexible number of reference views: **EscherNet**, evaluated on the **NeRF-Synthetic** dataset (Mildenhall et al., 2020) with 256px resolution; and **SEVA**, evaluated on the **LLFF** and **Mip-NeRF 360** datasets (Mildenhall et al., 2019; Barron et al., 2022) with 512px resolution.

In Fig. 7, we can observe that generative baselines such as EscherNet and SEVA initially outperform per-scene methods in few-view settings, their performance quickly plateaus. In contrast, Kaleido's performance continues to scale with more views, mirroring the trend of scene-specific methods. This superior zero-shot generalisation creates a widening performance gap over other generative models.

When all available reference views are used, Kaleido's performance on the NeRF-Synthetic dataset is nearly on par with Instant-NGP. On the more challenging LLFF and Mip-NeRF 360 datasets, Kaleido surpasses Instant-NGP, marking the first time a zero-shot generative model has matched the quality of a state-of-the-art, per-scene optimised method. Given that per-scene methods are often sensitive to camera coordinate systems, Kaleido's robust, data-driven performance highlights the immense potential of zero-shot solutions for general-purpose rendering. A detailed analysis of the computational trade-offs between generative and scene-specific methods is provided in Appendix C.

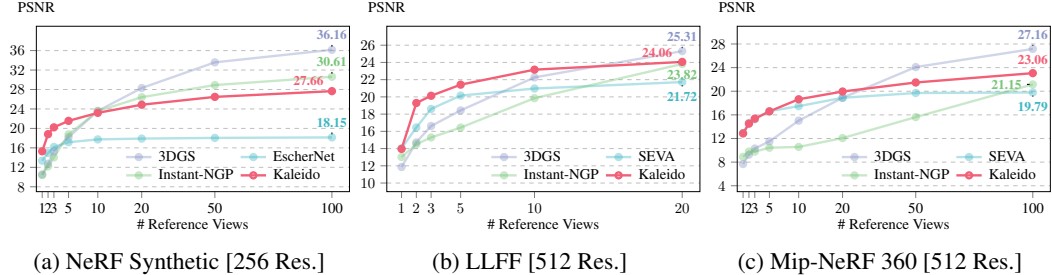

(a) NeRF Synthetic [256 Res.]    (b) LLFF [512 Res.]    (c) Mip-NeRF 360 [512 Res.]

Figure 7: **PSNR Performance with Per-Scene Optimisation Methods.** Kaleido's performance scales consistently with more reference views. It significantly outperforms other generative NVS baselines, with the performance gap widening as more views are provided. Notably, when given all available reference views, Kaleido surpasses Instant-NGP on both scene-level datasets.

## 4.2 RESULTS ON 3D RECONSTRUCTION

We evaluate Kaleido's 3D reconstruction capabilities on the **GSO-30** dataset by applying the **NeuS2** framework (Wang et al., 2023a) to Kaleido's generated views. We compare against a diverse set of image-to-3D baselines, including direct generation methods like **Point-E** (Nichol et al., 2022) and **Shape-E** (Jun & Nichol, 2023), optimisation-based methods like **DreamGaussian** (Tang et al., 2024), and other view-synthesis-based approaches like **One-2-3-45** (Liu et al., 2023a) and **Sync-Dreamer** (Liu et al., 2023d). Following the protocol of EscherNet, we generate a fixed set of 36 object-centric views to serve as input for the reconstruction. For a fair comparison, all methods are evaluated at 256px resolution, and we additionally report Kaleido's performance at 1024px to showcase its high-resolution capabilities.

Table 3: **3D Reconstruction Performance on GSO-30.** We measure reconstruction quality using Chamfer Distance (CD, lower is better) and Volumetric IoU (VIoU, higher is better). Kaleido clearly surpasses EscherNet by a large margin, demonstrating 5x greater view efficiency. The quality is further improved when using higher-resolution renderings from Kaleido.

| | 1 View | | 2 Views | | 3 Views | | 5 Views | | 10 Views | |
|---|---|---|---|---|---|---|---|---|---|---|
| | CD ↓ | VIoU ↑ | CD ↓ | VIoU ↑ | CD ↓ | VIoU ↑ | CD ↓ | VIoU ↑ | CD ↓ | VIoU ↑ |
| Point-E | 0.0447 | 0.2503 | – | – | – | – | – | – | – | – |
| Shape-E | 0.0448 | 0.3762 | – | – | – | – | – | – | – | – |
| One-2-3-45 | 0.0667 | 0.4016 | – | – | – | – | – | – | – | – |
| DreamGaussian | 0.0459 | 0.4531 | – | – | – | – | – | – | – | – |
| SyncDreamer | 0.0400 | 0.5220 | – | – | – | – | – | – | – | – |
| NeuS | – | – | – | – | 0.0366 | 0.5352 | 0.0245 | 0.6742 | 0.0195 | 0.7264 |
| EscherNet | 0.0314 | 0.5974 | 0.0215 | 0.6868 | 0.0190 | 0.7189 | 0.0175 | 0.7423 | 0.0167 | 0.7478 |
| **Kaleido** | 0.0214 | 0.6800 | 0.0120 | 0.7785 | 0.0113 | 0.7960 | 0.0104 | 0.8082 | 0.0100 | 0.8118 |
| **Kaleido [1024 Res.]** | 0.0183 | 0.7006 | 0.0118 | 0.7851 | 0.0104 | 0.8053 | 0.0091 | 0.8290 | 0.0086 | 0.8418 |

In Table 3, Kaleido again achieves the best performance in 3D reconstruction, outperforming direct image-to-3D models, our NeuS baseline, and EscherNet. The results highlight Kaleido's remarkable rendering efficiency and precision. With just 2 views, our model has surpassed the reconstruction quality that EscherNet achieves with 10 views. This superiority is more evident qualitatively in Fig. 10. Given the same 256px resolution, Kaleido's generated meshes are clearly better. At 1024px resolution, the reconstructed textures are incredibly detailed and sharp, appearing close to the ground truth and suggesting exciting new applications for high-fidelity, few-shot 3D reconstruction.

Finally, as Kaleido relies on a video data prior, it is conceptually related to camera-conditioned video generation models. We discuss these relationships and provide comparative results in Appendix H.

## 5 CONCLUSION

In this paper, we introduced Kaleido, a family of generative models that reframes neural rendering as a sequence-to-sequence problem. By gradually modernising the transformer architecture with a unified representation of space and time, Kaleido exhibits strong scaling and achieves state-of-the-art performance on view synthesis and 3D reconstruction benchmarks. Most notably, Kaleido is the first zero-shot generative model to match the quality of per-scene optimisation methods, marking a significant step towards a universal rendering engine. A discussion of the model's limitations and future work is included in Appendix I.

ETHICS STATEMENT

This research was conducted with a commitment to responsible AI practices. All data used to train our models was sourced from either publicly available academic datasets or was legally licensed from commercial providers, ensuring compliance with data usage rights. Our research was conducted with careful consideration for ethical principles, and to the best of our knowledge, it does not present issues related to privacy, security, legal compliance, or research integrity.

REPRODUCIBILITY STATEMENT

To ensure the reproducibility of our work, we have provided comprehensive details of our methodology. This includes detailed descriptions of our training data, evaluation strategies, and hyperparameters in Appendix E. A complete breakdown of our architectural design is provided in Section 3.2 and Appendix B.

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

## A  ADDITIONAL RELATED WORK

**From 2D to 3D and Camera Parameters**   Reconstructing 3D geometry and camera parameters from 2D images is a foundational problem in computer vision. Classical approaches like Structure from Motion (SfM) (Hartley & Zisserman, 2003; Schönberger et al., 2016) and Simultaneous Localisation and Mapping (SLAM) (Davison et al., 2007; Mur-Artal et al., 2015; Izadi et al., 2011) have been highly successful, but they are limited by their need to optimise each scene from scratch and their struggles with non-overlapping views. More recently, learning-based methods have emerged to address these limitations. Models like DUSt3R (Wang et al., 2024a) and VGGT (Wang et al., 2025) introduce feed-forward pointmap regression, enabling end-to-end 3D reconstruction that generalises across scenes. While these methods represent a significant step forward, their reliance on direct geometric regression means they cannot effectively infer content in occluded regions. Notably, Kaleido's fully generative design allows it to predict plausible, spatially consistent content for occluded regions, a key advantage over both classical and modern regression-based techniques.

**Multi-View Stereo, Neural Rendering, and Novel View Synthesis**   Traditional Multi-view Stereo (MVS) (Furukawa et al., 2015; Schönberger et al., 2016) reconstructs 3D surfaces by triangulating features across multiple viewpoints. This principle was revolutionised by Neural Radiance Fields (NeRF) (Mildenhall et al., 2020), which uses volume rendering and coordinate MLPs to achieve photorealistic novel view synthesis. A plethora of follow-up works have focused on improving the speed and quality of this per-scene optimisation paradigm (Müller et al., 2022; Fridovich-Keil et al., 2022; Chen et al., 2022; Kerbl et al., 2023). More recently, feed-forward transformer-based models have emerged (Hong et al., 2023; Jang & Agapito, 2024; Jin et al., 2024), which can directly predict 3D primitives or render novel views from limited inputs. However, as deterministic models, they fundamentally struggle with the inherently probabilistic nature of inferring occluded regions.

**Generative Video and World Models**   Kaleido's methodology is deeply connected to recent advancements in generative video and the emerging paradigm of world models. This progress is largely driven by a dominant technical stack combining diffusion or rectified flow models with transformer architectures, a foundation shared by many recent generative video model designs (Chen et al., 2025; Blattmann et al., 2023; Chen et al., 2024b;a; Yang et al., 2025). The increasing capabilities of video generation have also positioned it as a stepping stone towards building world models — systems that learn an internal model of the world to simulate physical interactions and predict future states. This trajectory is evident in models that focus on controllability and interactivity. For instance, the Navigation World Model (Bar et al., 2025) predicts future observations to facilitate planning, while frameworks like WonderWorld (Yu et al., 2025a) and GameFactory (Yu et al., 2025b) generate explorable 3D environments. Most notably, the Genie series (Bruce et al., 2024; Deepmind, 2025) creates interactive environments with persistent spatial memory and real-time promptable world events, marking a significant advance toward truly immersive and dynamic virtual worlds.

Kaleido contributes to this broader pursuit of world modelling from a different perspective. Instead of focusing on temporal dynamics or agent-based interactivity, Kaleido approaches world modelling through the lens of neural rendering, prioritising *spatial consistency* and *generation flexibility*. This unique approach allows it to operate across a spectrum of realities: with many reference views, it produces a grounded reality through faithful reconstruction; while with few views, it creates a generated reality with plausible unseen details. This unique capability to seamlessly transition between reconstruction and creative generation marks a distinct and intriguing path toward creating truly versatile and navigable virtual worlds.

# B  ADDITIONAL DETAILS ON KALEIDO DESIGNS

## B.1  UNIFIED POSITIONAL ENCODING DESIGN DETAILS

Our unified positional encoding represents different positions as follows: In 2D image positions, pixel coordinates are mapped to a pair of angles $(\theta_h, \theta_w)$, representing an element in $SO(2) \times SO(2)$, where $\theta_{h,w} \in [0, 2\pi)$ distributed uniformly from the top-left to the bottom-right patches; In temporal positions, frame indices are similarly mapped to a single angle $\theta_t \in SO(2)$, with values interpolated linearly from the start to the end of a clip. In 3D camera poses, 6-DoF camera extrinsics $c = \begin{bmatrix} \mathbf{R} & \mathbf{t} \\ 0 & 1 \end{bmatrix}$ (with rotation $\mathbf{R}$ and translation $\mathbf{t}$) are represented as an element in $SE(3)$, following the design in CaPE (Kong et al., 2024).

We can then define a unified geometric attribute $g$ for each token, depending on its data modality:

$$\text{For 3D data:} \quad g := (\theta_h, \theta_w, c) \in SO(2) \times SO(2) \times SE(3) \tag{3}$$

$$\text{For video data:} \quad g := (\theta_h, \theta_w, \theta_t) \in SO(2) \times SO(2) \times SO(2). \tag{4}$$

Within the GTA framework, these components are used to construct a block-diagonal transformation matrix $P_g$ that is applied to each token's feature vector $v \in \mathbb{R}^d$. The construction of $P_g$ varies for different attention blocks, allocating the feature dimension $d$ as follows: In Spatial Attention, we apply only the 2D position embeddings $(\theta_h, \theta_w)$, which are expanded into $d/4$ distinct frequency bands, with dimensions allocated to image height and width components based on a 1:1 ratio; In Temporal/3D Attention, the 2D embeddings $(\theta_h, \theta_w)$ are expanded into $d/8$ frequency bands. For video data, the temporal embedding $\theta_t$ is expanded into $d/4$ frequency bands. For 3D data, the pose embedding $c$ is repeated to fill the remaining dimensions. The total dimensions are allocated to image height, width, and temporal/3D components based on a 1:1:2 ratio.

Finally, we normalise the camera translation element $\mathbf{t}$ such that its maximum norm across all views in a given scene is 1. This ensures all positional transformations remain within a *bounded* range, which we found is crucial for stable training to handle different scene scales.

## B.2  RECTIFIED FLOW SNR SAMPLER DESIGN DETAILS

**Timestep Sampler for Training**    We evaluate three timestep sampling distributions, which were previously explored in SD3 (Esser et al., 2024), to assess their effect on learning efficiency and generalisation for neural rendering:

- **Uniform**: A standard uniform distribution $\mathcal{U}(0, 1)$.

- **Logit-Normal**: This distribution passes a value $u \sim \mathcal{N}(m, s)$ through a sigmoid function. The mean $m$ and standard deviation $s$ control the sampling bias. A positive mean ($m > 0$) focuses training on high-noise timesteps (closer to $t = 1$), while a negative mean ($m < 0$) focuses on low-noise timesteps (closer to $t = 0$).

- **Mode**: This distribution warps a uniform sample $u \sim \mathcal{U}(0, 1)$ with the mapping $1 - u - s \cdot (\cos^2(\frac{\pi}{2} u) - 1 + u)$. The parameter $s$ controls the sampling focus: $s > 0$ favors the midpoint ($t \approx 0.5$), while $s < 0$ favors the endpoints ($t \approx 0$ and $t \approx 1$).

We apply a modulation function, $m(t, \sigma) = \sigma \cdot t / (1 + (\sigma - 1) \cdot t)$, to skew each base distribution. A shifting factor $\sigma > 1$ pushes the probability mass towards the noise end of the trajectory ($t \approx 1$). The resulting probability densities are visualised in Fig. 8.

In **ablation (viii)**, our results showed that all distributions shifted towards the noise end significantly outperformed the unshifted baselines. While a heavily shifted uniform distribution performed marginally best, we chose **shifted Mode sampling** as our final design because it provides a superior balance between the critical early timesteps and the intermediate steps.

**Timestep Sampler for Inference**    In **ablation (ix)**, we introduce a **linear-quadratic sampling schedule** for inference to complement our training sampler. This schedule works as follows: for a total of $S$ denoising steps, the first $S/2$ timesteps are sampled linearly, while the remaining $S/2$ steps are sampled quadratically. This non-uniform schedule effectively concentrates more discrete denoising steps in the high-noise region of the trajectory, aligning the inference process with our training strategy and leading to substantial performance gains.

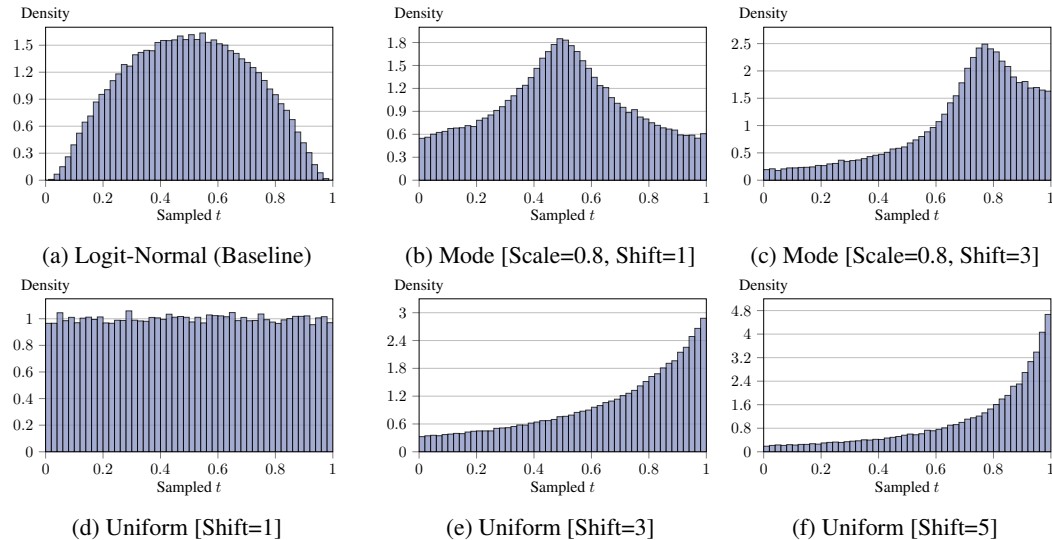

Figure 8: **Probability Densities of Different Timestep Sampler.** We visualise the PDFs for the samplers evaluated in our ablation study, where $t = 1$ is the noise end and $t = 0$ is the data end. (a) Logit-Normal: The standard baseline, which concentrates probability mass on the middle of the timestep range and has diminished density near the endpoints; (b) Mode: Similar to Logit-Normal, but maintains a positive density at the endpoints. (d) Uniform: A standard uniform distribution, which samples all timesteps with equal probability; (c), (e), and (f) show the Mode and Uniform distributions shifted towards the high-noise end to study the effect of noise-biased sampling.

## C  MEMORY CONSUMPTION AND RUNNING LATENCIES

In this section, we compare the memory consumption and inference latency of Kaleido against the generative and deterministic baselines evaluated in Section 4.1. In Table 4, the per-scene optimisation methods (Instant-NGP and 3DGS) operate in a distinct efficiency tier. They run significantly faster than generative approaches, as they do not require iterative denoising steps; 3DGS, in particular, achieves the fastest rendering speeds by not relying on the neural parameterisation entirely.

However, within the class of generative models, Kaleido demonstrates superior scaling efficiency. Notably, Kaleido-Medium is slightly larger than EscherNet in parameter count but scales much better with resolution (due to the efficient design of factorised spatial and temporal window attention), requiring only 10% of the rendering time on 1024 resolution. While our current design prioritises rendering quality and model scalability, narrowing the inference speed gap with deterministic, scene-specific methods remains an important direction for future work.

Table 4: **Memory Consumption and Inference Latency Comparisons.** We report the averaged per-frame rendering time for a sequence of 25 frames (1 reference + 24 target views) across both generative and deterministic methods at various resolutions. For generative models, the reported times reflect the full inference process based on the required denoising steps specified in the original papers. To evaluate memory efficiency, we additionally report the maximum number of frames for all generative methods that can be generated concurrently on a single GPU. All benchmarks were conducted on a single NVIDIA A100.

| | Generative (sec. per frames / max frames) | | | | | Deterministic (sec. per frame) | |
|---|---|---|---|---|---|---|---|
| | EscherNet [0.9B] (50 Steps) | SEVA [1.3B] (50 Steps) | Kaleido-Small [0.6B] (24 Steps) | Kaleido-Medium [1.2B] (24 Steps) | Kaleido [3.1B] (24 Steps) | InstantNGP | 3DGS |
| [256 × 256] | 0.58s / 1601 | 0.42s / 1473 | 0.55s / 231 | 1.1s / 201 | 2.7s / 161 | 0.35s | 0.0025s |
| [512 × 512] | 6.4s / 401 | 1.5s / 369 | 1.6s / 119 | 3.1s / 103 | 4.6s / 76 | 0.35s | 0.0025s |
| [1024 × 1024] | 92s / 101 | 8.6s / 93 | 5.4s / 55 | 10.0s / 46 | 17.6s / 35 | 0.67s | 0.0034s |

# D    KALEIDO DESIGN ABLATIVE QUANTITATIVE RESULTS

Table 5 presents the full quantitative results supporting the ablation study in Fig. 3, including all alternative design decisions we explored. To provide a comprehensive comparison, we report performance across three settings: one-to-five, five-to-one, and five-to-five reference-to-target views.

| | Objaverse | | | uCO3D | | | Training Throughput |
|---|---|---|---|---|---|---|---|
| | $1 \to 5$ | $5 \to 1$ | $5 \to 5$ | $1 \to 5$ | $5 \to 1$ | $5 \to 5$ | |
| Objaverse - Single Baseline | 14.56 | 19.53 | 21.23 | - | - | - | - |
| uCO3D - Single Baseline | - | - | - | 14.89 | 19.19 | 16.97 | - |
| Objaverse + uCO3D - Joint Baseline | 12.17 | 19.10 | 20.02 | 14.66 | 18.71 | 16.77 | 160 |
| **(i) Architecture Design [Vanilla DiT]** | | | | | | | |
| **DiT + Llama3 (SwiGLU + GQA)** | 13.02 | 20.18 | 21.23 | 14.63 | 19.31 | 17.27 | 160 |
| **(ii) Spatial Positional Encoding [2D RoPE + 3D CaPE]** | | | | | | | |
| RoPE + Plucker | 12.18 | 20.00 | 20.17 | 14.75 | 19.43 | 17.61 | 160 |
| **GTA [2D RoPE + 3D CaPE]** | 11.93 | 21.05 | 22.03 | 13.65 | 20.84 | 17.54 | 148 |
| **(iii) View Sampling Strategies [Fixed 6→6]** | | | | | | | |
| Uniform Sampling w/o Masking | 12.18 | 22.22 | 21.65 | 14.22 | 21.36 | 18.34 | 150 |
| Uniform Sampling w/ Masking | 14.51 | 21.00 | 20.22 | 14.58 | 20.37 | 17.60 | 150 |
| **Exponential Sampling w/ Masking** | 15.13 | 21.41 | 21.11 | 15.16 | 20.25 | 17.83 | 148 |
| **(iv) Temporal Attention Design [Temporal Attention (K=1)]** | | | | | | | |
| Full Attention | 14.54 | 22.62 | 22.40 | 15.00 | 20.75 | 18.28 | 58 |
| Temporal Window Attention (K = 2) | 15.45 | 21.60 | 21.04 | 15.40 | 20.43 | 18.15 | 146 |
| **Temporal Window Attention (K = 4)** | 15.67 | 22.25 | 21.79 | 15.55 | 20.81 | 18.45 | 142 |
| Temporal Window Attention (K = 8) | 15.73 | 22.60 | 22.54 | 15.85 | 21.39 | 19.15 | 103 |
| **(v) Auxiliary Features [None]** | | | | | | | |
| **DiNOv2 [DiT-B]** | 15.86 | 22.28 | 21.90 | 15.81 | 21.09 | 18.81 | 138 |
| DiNOv2 [DiT-L] | 16.34 | 22.65 | 22.43 | 15.94 | 21.24 | 18.75 | 135 |
| MetaDepth [DiT-L] | 15.82 | 22.28 | 22.22 | 15.76 | 21.26 | 18.65 | 135 |
| MetaNormals [DiT-L] | 15.77 | 22.32 | 22.11 | 15.59 | 21.20 | 18.85 | 135 |
| **(vi) Timestep Conditioning Design [AdaLN-Zero, Top 1 Act.: 15192]** | | | | | | | |
| Shift Only [Top 1 Act.: 6820] | 16.07 | 21.56 | 21.51 | 15.97 | 20.56 | 18.28 | 144 |
| No Timestep [Top 1 Act.: 4312.] | 16.26 | 20.85 | 21.00 | 16.15 | 20.38 | 18.38 | 148 |
| **(vii) Attention Registers [No Registers, Top 1 Act.: 15192]** | | | | | | | |
| **1 Register [Top 1 Act.: 397.75]** | 15.93 | 22.27 | 22.12 | 15.77 | 21.02 | 19.03 | 138 |
| 8 Registers [Top 1 Act.: 279.75] | 15.26 | 22.04 | 21.94 | 15.72 | 20.75 | 18.55 | 138 |
| 32 Registers [Top 1 Act.: 238.75] | 15.07 | 21.88 | 21.54 | 15.66 | 20.59 | 18.27 | 138 |
| **(viii) Timestep Sampling Training Strategy [LogitNorm [0,1]]** | | | | | | | |
| Uniform [Shift = 1] | 17.58 | 22.01 | 21.96 | 15.90 | 20.49 | 18.57 | 138 |
| Uniform [Shift = 3] | 18.27 | 23.64 | 23.28 | 16.39 | 21.74 | 19.21 | 138 |
| Uniform [Shift = 5] | 18.43 | 23.38 | 23.06 | 15.90 | 21.42 | 18.94 | 138 |
| Mode [Scale = 0.8] | 17.39 | 22.06 | 22.08 | 15.99 | 20.75 | 18.68 | 138 |
| **Mode [Scale = 0.8, Shift = 3]** | 18.19 | 24.06 | 23.75 | 16.03 | 21.76 | 19.11 | 138 |
| **(ix) Timestep Sampling Inference Sampling [Linspace [1, 999]]** | | | | | | | |
| Trailing [1, 980] | 17.95 | 23.66 | 23.51 | 16.70 | 21.83 | 19.43 | 138 |
| **LinearQuadratic [1, 999]** | 18.09 | 23.87 | 23.95 | 17.03 | 22.15 | 19.79 | 138 |
| **(x) with Video Pre-training [No video Pre-training]** | | | | | | | |
| Video Pre-training 100K Steps (1.3x Eff.) | 18.16 | 24.22 | 24.30 | 17.11 | 22.23 | 20.10 | 138 |
| **Video Pre-training 200K Steps (2x Eff.)** | 18.28 | 24.55 | 24.60 | 17.18 | 22.43 | 20.15 | 138 |

Table 5: **Quantitative Results for Kaleido Design Ablations.** We report the complete quantitative results (PSNR, higher is better) corresponding to the ablation study in Fig. 3. Performance is evaluated in one-to-five, five-to-one, and five-to-five reference-to-target view settings. Our final design choice for each component is marked in red.

# E   ADDITIONAL DETAILS OF KALEIDO TRAINING CONFIGURATIONS AND EVALUATION STRATEGIES

**Training Datasets**   Kaleido is trained on a diverse mixture of object-level and scene-level datasets. For object-level data, we use **ShutterStock 3D**, our licensed collection of synthetic 3D meshes, which we render with object-centric camera poses under varied lighting conditions; and **uCO3D** (Liu et al., 2025b), which includes real-world objects with estimated poses. For scene-level data, we combine several datasets: **RealEstate10K** (Zhou et al., 2018), which features indoor room scenes; **DL3DV** (Ling et al., 2024), which features both indoor and outdoor scenes; and a filtered subset of **ShutterStock Video**. This licensed video subset is curated to include only static scenes, and then labelled with a pose estimator VGGT (Wang et al., 2025).

In summary, our 3D fine-tuning dataset consists of approximately 1.5M object sequences and 2M scene sequences. For the initial video pre-training stage, we leverage the full, unfiltered Shutterstock Video dataset, comprising 34M video clips.

**Training Strategy**   Our training process follows a two-stage, progressive-resolution curriculum. First, we pre-train Kaleido exclusively on video data at a fixed 256px resolution. We then fine-tune on our combined multi-view 3D datasets, progressively increasing the resolution from 256px to 512px, and finally to 1024px. In our 1024px fine-tuning, we introduce multi-aspect-ratio training (including 1:1, 4:5, 5:4, 16:9, and 9:16) to enable flexible resolution generation.

All Kaleido model variants are trained using the same datasets, with the AdamW optimiser (Loshchilov & Hutter, 2019) and a weight decay of 0.01. In each training iteration, we randomly sample a total of 12 frames per sequence. These frames are then partitioned into reference and target views according to the view sampling strategy in Sec. 3.2.1.

The learning rate is chosen based on the training stage. For the initial video pre-training and the first stage of 3D fine-tuning (both at 256px resolution), we apply a learning rate of $10^{-4}$. For the subsequent high-resolution 3D fine-tuning stages (512px and 1024px resolution), we decrease the learning rate to $10^{-5}$. To train our larger models at large resolutions, we incorporate FSDP sharding and activation checkpointing.

We use `fp16` mixed-precision in all training stages, as we find it crucial for stable convergence; while `bf16` consistently leads to unstable training. Our largest Kaleido model is trained for two weeks on 512 NVIDIA H100 GPUs. Additional hyper-parameters are listed in Table 6.

| | Stage 1 (Video data) [$256 \times 256$] | | Stage 2 (3D data) [$256 \times 256$] | | Stage 3 (3D data) [$512 \times 512$] | | Stage 4 (3D data) [1024 mixed AR] | |
|---|---|---|---|---|---|---|---|---|
| | Batch Size | # Steps | Batch Size | # Steps | Batch Size | # Steps | Batch Size | # Steps |
| **Kaleido-Small** | 1024 | 700K | 1024 | 300K | 256 | 100K | 256 | 100K |
| **Kaleido-Medium** | 1024 | 700K | 1024 | 300K | 256 | 100K | 256 | 100K |
| **Kaleido** | 2048 | 700K | 2048 | 500K | 256 | 100K | 256 | 100K |

Table 6: **Kaleido Training Pipeline.** Kaleido's training follows a multi-stage curriculum. The model is first pre-trained on a large-scale video dataset and is then fine-tuned on combined multi-view 3D datasets, with the image resolution progressively increased from 256px up to 1024px. In the final stage, we sample images with mixed aspect ratios to enable flexible resolution generation. Larger batch sizes are used for our largest Kaleido model to validate scaling laws.

**Evaluation Strategy**   We evaluate Kaleido on both view synthesis (Section 4.1) and 3D reconstruction benchmarks (Section 4.2). All evaluation datasets were held out and not used during model training. For all experiments, we report the zero-shot performance of Kaleido without any per-dataset fine-tuning. Unless otherwise specified, all generations use a classifier-free guidance scale of 1.5. To ensure a fair comparison with prior work, the frame interpolation model is not used in these evaluations. In NVS benchmarking, we match our model's resolution to the baselines, using our 256px checkpoint against EscherNet (evaluated at 256px) and our 512px checkpoint against SV3D and SEVA (both evaluated at 576px). For single-view evaluations on scene-level datasets, we address scale ambiguity by sweeping camera translations along each and all axes (from 0.1 to 2.0) and reporting the best result, following the protocol of SEVA.

## F  ADDITIONAL RESULTS FOR NOVEL VIEW SYNTHESIS

We provide additional quantitative metrics for our few-view NVS benchmarks shown in in Table 2 and additional quantitative examples for our many-view NVS comparisons shown in Fig. 9.

| | OO3D | GSO-30 | | | | | RTMV | | | | | LLFF | | Mip-NeRF 360 | | | Tanks and Temples | | | |
|---|---|---|---|---|---|---|---|---|---|---|---|---|---|---|---|---|---|---|---|---|
| # Ref. Views | 1 | 1 | 2 | 3 | 5 | 10 | 1 | 2 | 3 | 5 | 10 | 1 | 3 | 1 | 3 | 6 | 1 | 3 | 6 | 9 |
| Eval. Data Type | Object | Object | | | | | Multi-Object | | | | | Scene | | Scene | | | Scene | | | |
| Eval. Resolution | 512 | 256 | | | | | 256 | | | | | 512 | | 512 | | | 512 | | | |
| Eval. Tar. Views | 20 | 15 | | | | | 10 | | | | | 5 | | 27 | | | 35 | | | |
| SoTA Model | SV3D | EscherNet | | | | | EscherNet | | | | | SEVA | | SEVA | | | SEVA | | | |
| Results (LPIPS↓) | 0.158 | 0.095 | 0.064 | 0.052 | 0.043 | 0.036 | 0.410 | 0.301 | 0.258 | 0.222 | 0.185 | 0.389 | 0.181 | 0.573 | 0.364 | 0.319 | 0.571 | 0.463 | 0.387 | 0.328 |
| **Kaleido-Small** | 0.144 | 0.123 | 0.061 | 0.043 | 0.029 | 0.019 | 0.332 | 0.204 | 0.166 | 0.130 | 0.095 | 0.323 | 0.152 | 0.528 | 0.376 | 0.318 | 0.549 | 0.449 | 0.385 | 0.328 |
| **Kaleido-Medium** | 0.126 | 0.094 | 0.048 | 0.034 | 0.023 | 0.015 | 0.329 | 0.181 | 0.145 | 0.109 | 0.080 | 0.315 | 0.127 | 0.473 | 0.347 | 0.290 | 0.508 | 0.437 | 0.359 | 0.302 |
| **Kaleido** | 0.121 | 0.086 | 0.044 | 0.030 | 0.021 | 0.013 | 0.289 | 0.171 | 0.137 | 0.105 | 0.074 | 0.301 | 0.123 | 0.530 | 0.344 | 0.286 | 0.541 | 0.465 | 0.363 | 0.288 |
| Results (SSIM↑) | 0.850 | 0.884 | 0.908 | 0.918 | 0.927 | 0.935 | 0.518 | 0.585 | 0.611 | 0.633 | 0.657 | 0.384 | 0.602 | 0.282 | 0.377 | 0.395 | 0.342 | 0.385 | 0.427 | 0.452 |
| **Kaleido-Small** | 0.873 | 0.867 | 0.919 | 0.938 | 0.954 | 0.969 | 0.584 | 0.670 | 0.703 | 0.746 | 0.800 | 0.341 | 0.574 | 0.221 | 0.313 | 0.362 | 0.313 | 0.359 | 0.403 | 0.444 |
| **Kaleido-Medium** | 0.880 | 0.885 | 0.933 | 0.948 | 0.963 | 0.975 | 0.591 | 0.697 | 0.731 | 0.778 | 0.827 | 0.359 | 0.645 | 0.271 | 0.347 | 0.410 | 0.351 | 0.359 | 0.419 | 0.459 |
| **Kaleido** | 0.884 | 0.895 | 0.938 | 0.954 | 0.966 | 0.978 | 0.610 | 0.704 | 0.738 | 0.781 | 0.836 | 0.375 | 0.659 | 0.248 | 0.361 | 0.433 | 0.333 | 0.368 | 0.429 | 0.479 |

Table 7: **Zero-shot SSIM/LPIPS Performance with Generative Methods.** Kaleido achieves state-of-the-art performance across all object- and scene-level benchmarks, with SSIM and LPIPS metrics consistent with the superior PSNR performance reported in Table 2.

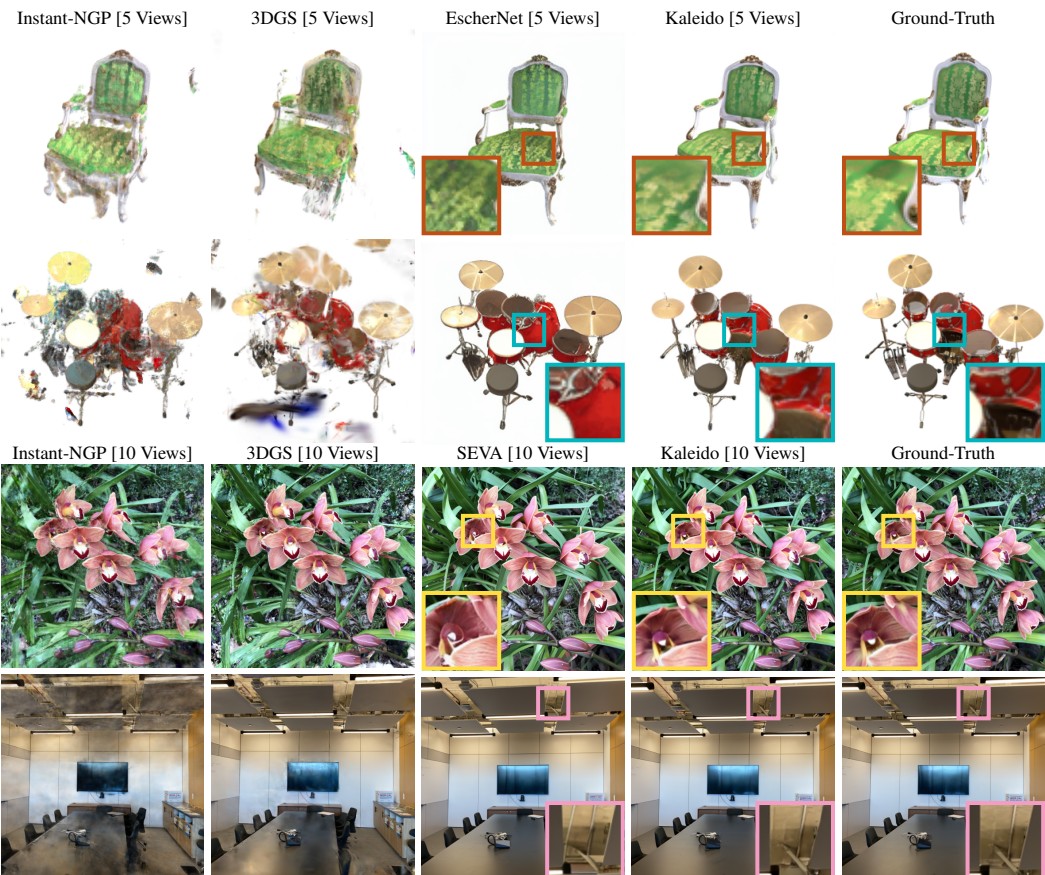

Figure 9: **Qualitative Comparison on NeRF-Synthetic (256px, top) and LLFF (512px, bottom).** With more reference views, Kaleido demonstrates superior rendering precision compared to other generative baselines, with more accurate texture details and pose alignment. Furthermore, it avoids the representation-based artefacts sometimes present in the per-scene optimisation methods, highlighting the robustness of its learned, data-driven prior.

# G ADDITIONAL VISUALISATIONS ON 3D RECONSTRUCTIONS

We include additional visualisation of our reconstructed object meshes in Fig. 10. The results show that Kaleido's generated views lead to significantly higher-quality meshes. Notably, at 1024px resolution, Kaleido's reconstructions is nearly close to the ground-truth, capturing fine geometric details and producing sharp, realistic textures.

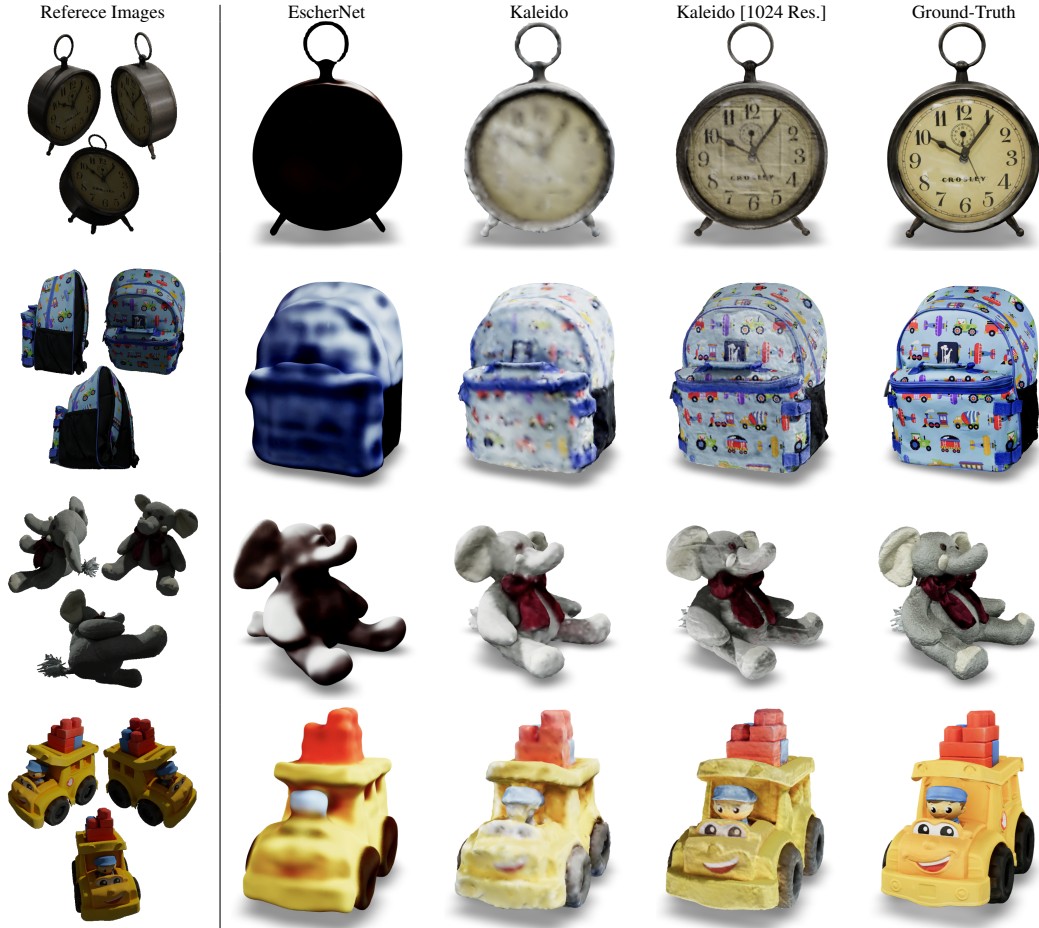

Figure 10: **Visualisation of 3D Reconstructions with 3 Reference Views.** Kaleido's precise renderings enable high-fidelity 3D mesh reconstruction using NeuS2. When leveraging 1024px renderings, the resulting meshes exhibit incredibly detailed textures, accurately capturing fine features like the numbers on the clock and the intricate patterns on the backpack.

# H  ADDITIONAL RESULTS FOR CAMERA-CONDITIONED VIDEO GENERATION

Kaleido leverages large-scale video data to improve novel view synthesis, a strategy that is conceptually related to recent camera-conditioned video generation models. While Kaleido is designed for the distinct problem of "any-to-any" spatial view synthesis, there is an overlap in capabilities when the task is restricted to a specific setting: generating a video sequence along a continuous target trajectory from a single reference view. In this section, we compare Kaleido against video generation models within this shared domain.

**Baselines and Datasets**  In Table 8, we compare Kaleido with a range of state-of-the-art camera-conditioned video models: **Wonderland** (Liang et al., 2025), **ViewCrafter** (Yu et al., 2025c), **VD3D** Bahmani et al. (2025), and **MotionCtrl** (Wang et al., 2024b). We evaluate on two challenging in-the-wild datasets: **DL3DV-140** (Ling et al., 2024) and **Tanks and Temples** (Knapitsch et al., 2017), strictly following the evaluation protocol described in Wonderland:

- **DL3DV-140**: We randomly select 300 video clips from the test set. For each video, the first frame serves as the reference image, and the subsequent $n$ camera poses are used as conditions. We set $n = 48$ to align with the Wonderland's setting.
- **Tanks and Temples**: We randomly sample 100 video clips from all 14 scenes using the same strategy.

Since both datasets lack dense pose annotations for every frame, we use COLMAP (Schönberger & Frahm, 2016; Schönberger et al., 2016) to generate ground-truth poses for the source videos.

**Evaluation Metrics**  We evaluate performance using two criteria, following Wonderland's setting:

- **Camera-Guidance Precision:** We measure **Rotation error** ($R_{err}$) **and Translation error** ($T_{err}$). To compute this, we estimate the camera poses of the generated videos using COLMAP, align the coordinate system relative to the first frame, and normalise the scale for comparison against the input condition.
- **Visual Similarity:** We assess image quality using **PSNR, SSIM, and LPIPS** between the generated frames and the ground-truth views. For reliable comparison, we evaluate these metrics over the first 14 frames. This is due to the generated content naturally diverges from the ground truth as the scene progresses, rendering pixel-wise metrics less indicative of generation quality for longer sequences.

**Results**  As shown in the Table 8, Kaleido clearly outperforms all camera-conditioned video models. The improvement is particularly significant in *camera precision*. Despite the baseline models being specialised for video generation along continuous trajectories, Kaleido has been shown to adhere to the target camera path with much higher accuracy. This highlights Kaleido's generality and superior rendering precision, even when applied to tasks outside its primary design scope.

Table 8: **Comparison with Camera-Conditioned Video Models.** We evaluate camera precision ($R_{err}$, $T_{err}$) and visual similarity (LPIPS, PSNR, SSIM) against state-of-the-art video generation baselines. Kaleido significantly outperforms all video models in camera precision, indicating superior geometric consistency, while maintaining visual quality on par with the state-of-the-art. This demonstrates Kaleido's robust generalisation, even compared to models specifically designed for continuous video trajectory generation.

| *Dataset* | Metrics | | | | |
|---|---|---|---|---|---|
| Method | $R_{err}\downarrow$ | $T_{err}\downarrow$ | LPIPS$\downarrow$ | PSNR$\uparrow$ | SSIM$\uparrow$ |
| *DL3DV-140* | | | | | |
| MotionCtrl | 0.467 | 1.114 | 0.309 | 14.35 | 0.385 |
| VD3D | 0.094 | 0.237 | 0.259 | 16.28 | 0.487 |
| ViewCrafter | 0.092 | 0.243 | 0.237 | 17.10 | 0.519 |
| Wonderland | 0.061 | 0.130 | 0.218 | 17.56 | 0.543 |
| **Kaleido** | 0.011 | 0.026 | 0.232 | 18.09 | 0.544 |
| *Tanks and Temple* | | | | | |
| MotionCtrl | 0.834 | 1.501 | 0.312 | 14.58 | 0.386 |
| VD3D | 0.117 | 0.292 | 0.284 | 15.35 | 0.467 |
| ViewCrafter | 0.125 | 0.306 | 0.245 | 16.20 | 0.506 |
| Wonderland | 0.094 | 0.172 | 0.221 | 16.87 | 0.529 |
| **Kaleido** | 0.016 | 0.086 | 0.197 | 17.97 | 0.528 |

# I   LIMITATIONS AND FUTURE WORK

Despite its strong performance, Kaleido has several limitations that open exciting avenues for future research:

**Fixed Camera Intrinsics**   Kaleido currently does not model camera intrinsics, which prevents it from generating effects like dolly-zooms, a capability present in models like SEVA (Zhou et al., 2025). Future work could explore incorporating intrinsic parameterisation, potentially through another form of RoPE-based positional encoding designs (Li et al., 2025), to allow for more flexible camera control.

**Degraded Generations with Large Viewpoint Changes**   While Kaleido maintains excellent spatial consistency, its generated views can sometimes lack semantic plausibility when the viewpoint change is extreme. This suggests that while video pre-training builds a strong geometric foundation, it may not provide the diverse semantic knowledge required for high-fidelity single-image realism. Integrating priors from large-scale text-to-image/video models could be a promising direction to address this limitation.

**Towards Faster Rendering**   Kaleido's generation time scales with the number of input views, and it is far from real-time. To fully bridge the gap with efficient scene-specific methods like 3D Gaussian Splatting, future work will focus on improving inference speed through techniques like step distillation or architectural optimisations.

**Towards 4D Generation**   Our unified positional encoding for space and time provides a natural foundation for true 4D generation. A promising future direction is to extend Kaleido to precisely control scenes across both space and time, enabling generative modelling of dynamic, four-dimensional worlds.

