# OpenReview forum: "Scaling Sequence-to-Sequence Generative Neural Rendering"
_ICLR.cc/2026/Conference — ICLR 2026 Poster_

### Official Review · Reviewer_7YYk · 2025-10-15

**Soundness:** 2
**Presentation:** 2
**Contribution:** 2
**Rating:** 2
**Confidence:** 5

**Summary:**

This paper proposes Kaleido, a diffusion transformer for neural rendering. The model is jointly trained on 2D video and 3D data to take image(s) and output images from novel viewpoints. The paper proposes a new positional encoding to facilitate joint training on 2D video and 3D data.

**Strengths:**

- New positional encoding layer: This work combines RoPE for 2D video and 3D to jointly train on both data modalities.

**Weaknesses:**

- Limited viewpoint changes: The synthesized videos don’t show very large viewpoint change or show any synthesized “new” content not available in the input. It seems like the method requires many images to increase the viewpoint change. Though, methods like CAT3D can synthesize large viewpoint changes from a single image. Moreover, recent camera-controlled video models can synthesize very long trajectories.
- Flat geometry: Most generated trajectories have very flat geometry without good depth.
- Missing comparisons with camera-controlled video diffusion: This work looks conceptually similar to camera-controlled video diffusion models but comparisons with those kind of works are missing and the key differences in the approach are not clear. Moreover, in that case it would be possible to just use MegaSaM [1] or ViPE [2] to annotate all 2D videos with camera poses and directly train on large-scale video data for novel view synthesis.
- Simple datasets: The model is trained with limited scale 3D datasets. Is it a current issue that the model is not cross-generalizing from the synthetic 3D datasets to the video dataset distribution?

[1] Li et al., MegaSaM: Accurate, Fast, and Robust Structure and Motion from Casual Dynamic Videos, CVPR 2025 \
[2] Huang et al., ViPE: Video Pose Engine for 3D Geometric Perception, arXiv 2025

**Questions:**

I am not very convinced by the paper. It makes a big deal out of jointly training on 2D and 3D data, but it is not clear what the advantage or novelty of this work is. There are many camera-controlled video diffusion models that can just be used for joint fine-tuning. The architecture of Kaleido looks very similar to that of a regular diffusion transformer.

I would like authors to address the following questions:

- How does this compare to camera-controlled video diffusion models qualitatively and quantitatively? And why would someone not just jointly fine-tune a video model on 2D video data and multi-view data?
- Why are the viewpoint changes limited so much?
- Why is the geometry often flat?

I am not very optimistic that I can be convinced to accept this paper, since it is missing critical comparisons and the results are not very convincing. Moreover, the storyline is not convincing to motivate why we need this from-scratch-trained model. But still happy to see a rebuttal.

---

> ### Author Response · Authors · 2025-11-18
> **Rebuttal**
>
> We sincerely thank Reviewer 7YYk for their time and detailed feedback. We respectfully believe there has been a fundamental misunderstanding of our core task, which has led to incorrect assumptions about our contributions and results. We will address each point in detail.
>
> ### **1\. On 3D Generative Rendering v.s. 4D Video Generation**
>
> The reviewer's primary confusion seems to stem from the assumption that Kaleido is a "camera-controlled video diffusion model." **This is incorrect.**
>
> * **Kaleido is a 3D Generative Neural Rendering or Novel View Synthesis Model.** The goal is to generate a *spatially consistent* novel view from **any arbitrary 6-DoF pose**, conditioned on **any number** of reference views. This is a flexible "any-to-any" spatial query problem.
> * **Camera-controlled video generation is a 4D temporal prediction problem.** The goal is to generate a *temporally consistent* video that follows a *continuous* camera path. These models are not designed to handle arbitrary 6-DoF spatial queries from an arbitrary set of N reference views.
>
> This distinction is critical. "just fine-tuning a video model" is not a viable solution, as these models lack the fundamental "any-to-any" design features that Kaleido introduces. Kaleido were not jointly trained on video and 3D at the same time. Our use of video is to pre-train a *spatial prior* ("visual commonsense"), not to perform temporal prediction.  We have sharpened this distinction in our related works.
>
> ### **2\. On Flat Geometry**
>
>
> This claim is factually incorrect. We ask the reviewer to please re-examine **Table 3 in our paper  and the generated 3D assets in our demo page**, where we show **SoTA 3D reconstruction results**. Our model's renderings produce meshes with 1.83 Chamfer Distance and 70%VIoU, conditioned on only a single frame, significantly outperforming all prior work. It is *impossible* to achieve this state-of-the-art 3D reconstruction from "flat" renderings.
>
> ### **3\. On "Limited Viewpoint Changes"**
>
> We respectfully disagree with this claim and apologize for not sign-posting our most compelling results more clearly.
>
> * **Existing Demos:** Our original supplementary page already included **720-degree spiral trajectories** (NeRF-Synthetic) and a **480-frame autoregressive generation** of the Mip-NeRF "Room" scene, which features a long, complex, looping trajectory.
> * **New Demos:** To further address this concern, **we have added new single-view conditioned 360-degree generation demos** for both objects and scenes to our project website. Crucially, the object demos use the **exact same view inputs as CAT3D**. This direct comparison shows Kaleido's strong performance on large viewpoint changes. Furthermore, we would like to highlight that Kaleido's results are **single-pass**, whereas CAT3D's outputs are inconsistent and require a second-stage SDS refinement.
>
> ### **4.. On "Simple Datasets" and "Cross-Generalization"**
>
> We disagree with the characterization that our model's generalization is limited. As shown extensively on our project website, Kaleido demonstrates **incredible zero-shot generalization** to *highly* out-of-distribution, in-the-wild images, including **impressionist paintings, reflective surfaces, and complex materials**.
>
> Furthermore, Kaleido shows unique **emergent capabilities** by successfully rendering unconventional inputs like **image collages and padded images**, a capability not shown in any prior rendering model. This is the advantage of our data-driven, video-pre-trained approach.
>
> ### **5\. On MegaSaM / ViPE**
>
> We thank the reviewer for the suggestion. Ironically, our own internal (unrelated) experiments with ViPE have shown that its pose estimations can be extremely noisy on in-the-wild videos, making it unsuitable for training a high-precision rendering model like Kaleido. And again, ViPE and MegaSAM are designed for dynamic scene pose estimation, aiming for 4D modelling which is different from Kaleido’s design objective. The challenge of 4D modelling is also precisely why learning a foundational prior from unlabeled video is a valuable and challenging research direction.
>
> We hope these clarifications and our new 360-degree demos, which directly address the reviewer's main concerns, will prompt a re-evaluation of our work's novelty and significant contributions.

---

### Official Review · Reviewer_xBaB · 2025-10-30

**Soundness:** 3
**Presentation:** 3
**Contribution:** 2
**Rating:** 6
**Confidence:** 3

**Summary:**

This paper presented a method to generate novel view images from input views using the sequence-to-sequence generative model. They conducted numerous validate experiments to design the final model, such as the positional encodding, model architecture, and video pretraining, which can achieve the certain consistent view synthesis from both sparse and many view conditions.

**Strengths:**

1. The model design of this paper is motivated by many ablation studies. This paper designed the model mainly from five aspects, including the positional encodding, activation function and so on, and most designs have the corresponding ablation studies.
2. This paper conducted both 3D reconstruction and novel view synthesize comparisons with existing methods and show its advantage.
3. The results with many-view setting show that it can achieve more consistent results compared with existing methods.

**Weaknesses:**

1. Adopting the generative model to synthesis the novel view images is a common solution and is just one posible solution to achieve the final reconstruction of 3D object or scene. It has clear advantage in challenging situation like generating the non-seen views, but on the contraty, it has clear shortcoming in the consistency rendering results and efficiency.
2. To my knowledge, many previous methods have treated the multi-view synthesis as the video generation, and they can also generative any number of target views conditioned by any number of input views. And the declared novelty in this field is not clear to me.
3. In the many-view setting, seems like luck the experiments with long sequence where the inputs not just have numerous views but have long trajectory like SEVA.
4. Most of the technologies used may not seem novel, but the key is to choose the right combination.

**Questions:**

Missing the information of the memory and runtime comparisons with both previous generative methods and reconstruction methods like 3DGS, and how many input views can be processed at most and the corresponding runtime time?

---

> ### Author Response · Authors · 2025-11-18
> **Rebuttal**
>
> We sincerely thank Reviewer xBaB for their positive assessment and for recognizing our thorough ablation-driven design and strong quantitative results.
>
> To answer each of the reviewer’s questions:
>
> 1. The reviewer makes an excellent point about the central trade-off in 3D research. We agree that, historically, generative models have lagged behind per-scene methods in consistency and efficiency. **Our paper's primary goal is to challenge this assumption and closing the performance gap.** Kaleido is the **first zero-shot generative model to match the quantitative performance of SoTA per-scene-optimized methods like InstantNGP**. We hope these results show that the trade-off is no longer just *quality vs. generalization*, but *per-scene optimization vs. zero-shot inference*. Kaleido can render a *new* scene in seconds, whereas optimization-based methods require minutes with *per-scene training*. We believe our scaling recipe, which pushes generative quality to this new level, is a significant contribution.
> 2. We thank the reviewer for this point and will clarify this in our updated related works. As detailed in **General Comment 1**, there is a crucial distinction between our task (**3D rendering, NVS**) and **4D camera-controlled video generation**. To our knowledge, no prior work has used video pre-training for a scalable, **seq-to-seq "any-to-any" 3D NVS model** as we do. This specific "video-to-3D" transfer of a *spatial prior* is a core part of our novel contribution. We would be grateful if the reviewer could directly provide any references to works they believe we have missed that share this design and motivation.
> 3. We thank the reviewer for this suggestion. In fact, **we already have this exact comparison in our supporting demo website**, which we will signpost more clearly. Our MipNeRF 'Room' demo shows a **480-frame, 720-degree rotation** conditioned on just 8 images. This is an autoregressive "stress test" (40x longer than our training data) that directly demonstrates long-trajectory stability. We would also note that when comparing to public demos from SEVA with 8-view conditioning, SEVA exhibits significantly more flickering, highlighting the strength of our model's learned prior.
> 4. We acknowledge but disagree with the reviewer's comment: "the key is to choose the right combination." Our paper is not about finding the  final combination, but about the **systematic ablation study that *identifies* the non-obvious, unique scaling bottlenecks** that have prevented a simple seq-to-seq model from working before, making 3D significantly lacking behind from video and image generation. Our novel contributions are the both *identification and solution* for these bottlenecks, such as the **massive activation overflow** (Sec 2.2.2) and the **suboptimal SNR samplers** (Sec 2.2.3), which are unique to scaling generative *rendering* models. This "scaling recipe" itself is a core contribution.
> 5. We agree this is a crucial comparison. As detailed in **General Comment 4**, we have **added a new table and discussion to the appendix C** comparing the runtime and memory of Kaleido, 3DGS, and SEVA, clarifying the practical trade-offs of our zero-shot approach.
>
> We hope these clarifications and our new appendix section have fully addressed the reviewer's concerns.

---

> > ### Author Response · Authors · 2025-11-26
> >
> > Dear Reviewer xBaB,
> >
> > We wanted to follow up to ensure you have had a chance to review our response. We hope we have resolved your concerns regarding the model's contributions and scope. We are happy to answer any further questions you may have.

---

### Official Review · Reviewer_Q5DW · 2025-11-01

**Soundness:** 3
**Presentation:** 3
**Contribution:** 2
**Rating:** 6
**Confidence:** 4

**Summary:**

This paper presents a sequence-to-sequence method to achieve multi-view image synthesis, and they construct the pipeline from the positional encoding to the video pretraining. The final results show that this model has advantage in both single or multiple view input.

**Strengths:**

Strengths:

- Proposed the unified positional encoding to seamlessly represent multi-view and temporal positions.

- The presentation structure is reasonable and most design modules have corresponding ablation studies.

- The results with single or multiple view inputs show the advantage in both 3D and novel view synthesis.

**Weaknesses:**

Weakness:

- Missing the discussion on the running time and memory consumption (especially with the classical reconstruction representations like 3DGS). This kind of generation methods has the advantage on the generation of the nun-seen part, but it still needs huge memory consumption and long running time.

- From the proposed demos, the generated results still have obvious ghosting and artifacts (e.g., the generation attempts at the same location is inconsistent), this is still the inherent disadvantage of this kind of generation methods.

- How about the results when the input views and the target views have large difference in perspective? Maybe this challenging situation can further prove the effectiveness.

**Questions:**

As mentioned in the Strengths and Weaknesses, I am inclined to give a borderline-accept score; however, I would be happy to raise it if the authors address these main concerns.

---

> ### Author Response · Authors · 2025-11-18
> **Rebuttal**
>
> We sincerely thank Reviewer Q5DW for their positive assessment and for recognizing the strengths of our unified positional encoding, our thorough ablations, and our strong performance.
>
> To answer each of the reviewer’s questions:
>
> 1. We completely agree that this is an important comparison. As detailed in our **General Comment 4**, we have added a **new section to the Appendix C** that provides a detailed breakdown of Kaleido's inference latency and memory consumption, comparing it directly to both generative baselines and per-scene optimization methods. We hope these new results clarify the practical trade-offs of our zero-shot paradigm.
> 2. We thank the reviewer for this observation. We believe this comment may refer to our 480-frame "Room" demo, which, as noted in **General Comment 3**, is an extreme stress test generated autoregressively for **40x longer** than our training sequence length. We would kindly ask the reviewer to also consider our other single-view generation demos (e.g., the out-of-distribution generation on Monet and Dali paintings) which demonstrate exceptional stability. More importantly, our **SOTA quantitative metrics (Table 2\)**, which show a significant lead in PSNR, provide objective evidence that Kaleido is *quantifiably more consistent* and produces *fewer artifacts* than prior generative methods.
> 3.  To further reinforce Kaleido's capability here, **we have added new single-view conditioned 360-degree generation results** to our project website. These new demos use the *exact same single-view inputs* from other works (like CAT3D and GEN3C) and directly showcase Kaleido's strong performance on extreme viewpoint changes.
>
> We hope these clarifications, along with the new results and appendix section, have fully addressed the reviewer's concerns.

---

> > ### Comment · Reviewer_Q5DW · 2025-11-26
> > **Official Comments by Reviewer  #Q5DW**
> >
> > Thanks for the response of authors. And after checking the comments of other reviewers and the feedback, I have no further issues. I keep my initial score.

---

> > > ### Author Response · Authors · 2025-11-26
> > >
> > > Dear Reviewer Q5DW,
> > >
> > > We appreciate you checking our rebuttal and confirming that all issues are resolved.
> > >
> > > Given that these main concerns have been addressed, we kindly hope you might reconsider adjusting the rating to reflect the improved completeness and quality of the work, as mentioned in your initial review.
> > >
> > > Best regards,

---

### Official Review · Reviewer_F7xZ · 2025-11-01

**Soundness:** 2
**Presentation:** 3
**Contribution:** 2
**Rating:** 4
**Confidence:** 4

**Summary:**

Method Kaleido solved the object- and scene- level synthesis through the sequence-to-sequence generation technology. And they mainly borrowed the video generation model to achieve this and assisted some structural designs like activation function to improve the performance.

**Strengths:**

- The results show that this method have advantage in both few- and many- view settings, and they can match the performance of classical 3D representations under the many-view setting.

- This paper explored many detailed model designs including the positional encoding and architecture designs through ablation studies.

**Weaknesses:**

- Adopting the video generation to achieve novel view synthesis is a common solution, and many methods still adopt the video pretrained model to improve the performance. So the contribution on the video pretrained model is unconvincing.

- Once this model unified the 3D and video, so how about the performance on the 4D scene.

- The demo results still have obvious inconsistency in the generated novel view.

**Questions:**

N/A

**Details Of Ethics Concerns:**

There are not obvious ethics issues.

---

> ### Author Response · Authors · 2025-11-18
> **Rebuttal**
>
> We thank Reviewer F7xZ for the time and for acknowledging our strong results in both few- and many-view settings, as well as our extensive ablation studies.
>
> To answer each of the reviewer’s questions:
>
> 1. We respectfully disagree that this is a "common solution." As we detail in **General Comment 1**, there is a *fundamental* difference between our task (**3D Generative Neural Rendering**) and **Camera-Controlled Video Generation**:
>    * Kaleido is designed to tackle the "any-to-any" 6-DoF *view synthesis* problem.
>    * Camera-conditioned video generation models are designed to tackle the *temporal* prediction problem, often with heavy constraints (e.g., text-to-video, or image-to-video from a single start frame).
>    * Our novelty is in using video to learn a *spatial prior* ("visual commonsense") for 3D, enabled by our unified positional encoding. This specific pre-training paradigm for a scalable, seq-to-seq *rendering* model is novel. We would be grateful if the reviewer could directly provide any references to works they believe we have missed that share this design and motivation.
> 2. This relates to the core misunderstanding above. Kaleido is a **3D generative model** for static scenes. It is not designed to be a 4D generation model. Our unified representation allows the *architecture* to process 1D time (from video) and 6-DoF poses (from 3D) within the similar parameter-free positional encoding design, but the *goal* is not 4D output. As highlighted in **General Comment 1**, 4D generation is a different and non-comparable research problem.
> 3. We believe this comment refers to our 480-frame "Room" demo. As we clarify in **General Comment 3**, this video is an extreme **stress test**, generated autoregressively for **40x longer** than the 12-view sequences seen during training. We would ask the reviewer to also consider our other demos (e.g. our out-of-distribution generations on Monet and Dali paintings) and our **newly added 360-degree generations.** Our SoTA quantitative metrics (Table 2 in our paper) also confirm that Kaleido archives significantly better rendering precision than any prior generative methods.
>
> We hope these clarifications, combined with our General Comments, help to resolve the reviewer's concerns about our novelty and contribution. We are happy to discuss this further.

---

> > ### Author Response · Authors · 2025-11-26
> >
> > Dear Reviewer F7xZ,
> >
> > We wanted to follow up to ensure you have had a chance to review our response. We hope we have resolved your concerns regarding the model's contributions and scope. We are happy to answer any further questions you may have.

---

### Author Response · Authors · 2025-11-18
**General Comments (Part 1/3)**

We sincerely thank all reviewers for their time and thoughtful feedback on our work. We are encouraged by the reviewers’ recognition of our strong quantitative results, our extensive ablations, and our model's ability to match per-scene optimization methods.

Below, we first address shared concerns raised by multiple reviewers with general comments aimed at resolving these issues. Following this, we provide detailed, individual responses to each reviewer, outlining **our proposed and updated paper changes (marked as colour orange)** to address their specific feedback.

## **Kaleido Problem Definition \[F7xZ, XBaB, 7YYk\]**

We would like  to clarify a crucial point. There appears to be a misunderstanding of our core task, which is **Generative Neural Rendering / Novel View Synthesis  (NVS), not Camera-Controlled Video Generation**. These are two distinct research fields with different goals, and we have clarified this distinction in our updated related works.

**Camera-Controlled Video Generation** aims to inject camera control into **temporal prediction**. The goal is to generate a *temporally consistent video* that follows a specific camera *path*. These models are inherently 4D (3D space \+ 1D time) and are typically designed for continuous trajectories with specific constraints:

* **MotionCtrl \[SIGGRAPH 2024\] / CameraCtrl \[ICLR 2025\]:** These are text-to-video pipelines and do not use reference images as required by NVS models.
* **CameraCtrl II \[ICCV 2025\]:** This is an image-to-video pipeline, but it is constrained to a *single* reference image, which must also be the *starting frame*.
* **ReCamMaster \[ICCV 2025\]:** This model requires the reference and target sequences to be of the *same temporal length*.

**Generative Neural Rendering (Kaleido's objective)**, in contrast, is a **3D problem**. The goal is to generate *spatially consistent* novel views from **any arbitrary 6-DoF pose**, conditioned on **any number of reference views**. The target pose has no required temporal relationship to the inputs; it is a flexible "any-to-any" spatial query.

Because Kaleido is designed to tackle  this "any-to-any" 3D spatial problem, it is fundamentally not comparable to 4D models. Our use of video data is solely for *pre-training a spatial prior*, not for temporal prediction.

---

> ### Author Response · Authors · 2025-11-18
> **General Comments (Part 2/3)**
>
> ## **Kaleido Contributions**
>
> We believe the reviewers have partially overlooked Kaleido contributions by equating our approach with "common" video generation.
>
> First, Kaleido is a **generative rendering model** that can synthesize **any number of target views** from **any number of reference views** with **any 6-DoF camera pose**. This flexible "any-to-any" spatial query capability is a fundamental design goal that, to our knowledge, no camera-controlled video model can achieve.
>
> Second, our work modernizes a field that has historically lagged behind image and video generation, particularly on scalable architecture designs and generation qualities. Before Kaleido, the top generative rendering models were built on **U-Nets** with pre-trained image priors (e.g., **SEVA**, **CAT3D**, **ReconFusion**, **ZeroNVS**), which are difficult to scale and are limited to **\~512/576px resolution**. Many of these (**CAT3D**, **ReconFusion**, **ZeroNVS**) also require **multi-stage SDS refinement** to achieve consistency. Even video-prior methods like **ViewCrafter** still rely on explicit 3D representations (3DGS for consistent multi-view rendering).
>
> Kaleido is the **first model** to successfully combine several key components that, together, solve the unique scaling challenges of generative rendering **without any 3D-specific architectural design and representations**:
>
> 1. It is the first generative rendering model built on a **modern diffusion transformer** (using rectified flow) that is **pre-trained on videos then efficiently fine-tuned on 3D**  using a unified, shared architectural design.
> 2. It is the first generative rendering model to achieve **photorealistic 1024px resolution**, moving beyond the 512/576px limitations of prior work (like SEVA, CAT3D).
> 3. It is the first generative rendering model to **match the performance of per-scene optimization methods like InstantNGP** in many-view settings, and it does so **without relying on any SDS-based refinement** (which is a  common requirement by many prior methods: CAT3D, ZeroNVS, ReconFusion).
>
> We want to also highlight how Kaleido's design is more scalable and advanced than other recent SoTA generative rendering models:
>
> * **SEVA \[ICCV 2025\]:** While SEVA was arguably the strongest general-purpose NVS model before Kaleido, it is built on an pre-trained image model with a U-Net architecture, which is known to be less scalable than a pure transformer and is bounded at 576px resolution. Kaleido’s design is simpler and scales more effectively, and has shown to have much less flickering artefacts (comparisons available at the demo website: \[[https://stable-virtual-camera.github.io/](https://stable-virtual-camera.github.io/)\]).
> * **RTFM \[World Labs, released in Oct, 2025\]:** This model was released *after* the ICLR submission deadline produced by a superstar commercial lab. A comparison of its blog post \[[https://www.worldlabs.ai/blog/rtfm](https://www.worldlabs.ai/blog/rtfm), under Section \- Scalability: World Models as Learned Renderers\] and Kaleido demo videos clearly shows that Kaleido’s visual quality is significantly higher, with far fewer flickering artifacts and support for non-square aspect ratios.

---

> > ### Author Response · Authors · 2025-11-18
> > **General Comments (Part 3/3)**
> >
> > ## **Small View Change / New Generations \[Q5DW, 7YYk, xBaB\]**
> >
> > Our demos showing small view changes (e.g., on impressionist paintings, etc) were intended to highlight Kaleido's exceptional **generative consistency** and **zero-shot generalization**. Unlike prior models such as CAT3D \[[https://cat3d.github.io](https://cat3d.github.io)\], EscherNet  \[[https://kxhit.github.io/EscherNet](https://kxhit.github.io/EscherNet)\], and ReconFusion  \[[https://reconfusion.github.io](https://reconfusion.github.io)\] , which often produce strong flickering artifacts that necessitate a second-stage SDS refinement, Kaleido generates these ultra-smooth, high-quality videos in a **single pass, with no post-processing**. This demonstrates a significant improvement in generative stability.
> >
> > We apologize for not sign-posting our large-view-change capabilities more clearly. These results **were already included in our original submission**:
> >
> > * **NeRF-Synthetic with flexible input views:** features a continuous **720-degree spiral** trajectory.
> > * **Mip-NeRF 360 "Room" scene:** demonstrates our model's upper bound by **autoregressively generating 480 frames** (40x more than our 12-view training limit) on a complex, long-trajectory path.
> >
> > To further address this concern, **we have added new 360-degree generation videos** to our project website. These new results use the **same input views as CAT3D and GEN3C**, allowing for a direct and fair comparison of large-viewpoint synthesis. (Note: Due to memory constraints, these are generated jointly in 48 views per scene).
> >
> > -> Check New Demos here: \[[https://kaleido-research.github.io/](https://kaleido-research.github.io/)\]
> >
> > ## **Memory Consumption \+ Running Latencies \[Q5DW, xBaB\]**
> >
> > We agree that a direct comparison of computational cost is crucial for understanding the trade-offs between our zero-shot generative paradigm and per-scene optimization methods.
> >
> > While our current work prioritizes generative quality, scalability, and zero-shot flexibility, we recognize that inference efficiency is a key factor. To address this, **we have added a new section to the Appendix C** comparing Kaleido's memory usage and inference latency with both generative baselines and per-scene methods.
> >
> > This new data clarifies the expected trade-offs: Kaleido's zero-shot inference, while not real-time, allows it to render *any new scene directly*. This is a fundamental advantage over per-scene methods, which require a costly (minutes-long) optimization phase for every new scene. We agree and have acknowledged in our paper Limitation Section that closing this inference gap is an important direction for future work.

---

> > > ### Comment · Reviewer_7YYk · 2025-11-19
> > > **Multi View to 3D**
> > >
> > > Thanks for the response. How does this work compare with camera-controlled video generation works that can go from a sparse set of input views to a synthesized video? E.g., GEN3C (https://research.nvidia.com/labs/toronto-ai/GEN3C/) has results on their website with 5 input images. For me you can always just fine-tune a video model to take more conditioning views, since if the model works with one input view, it will also work just better with more input views.

---

> > > > ### Author Response · Authors · 2025-11-19
> > > > **Response to 7YYk**
> > > >
> > > > We thank the reviewer for the quick follow-up. We would like to use this opportunity to further clarify the technical distinctions between Kaleido, standard video models, and GEN3C.
> > > >
> > > > **1. Why "Just Fine-Tuning" Video Models is Non-Trivial**
> > > > The reviewer suggests that "if a model works with one view, it will work better with more." While this intuitively applies to **3D architectures**, it does not apply to **video architectures**.
> > > > Most camera-controlled video models (e.g., CameraCtrl II, described above) are architecturally constrained to accept a single reference image specifically as the starting frame.
> > > > * To accept any number of reference views at arbitrary spatial positions, the architecture must be able to encode the **relative 3D transformations** between all input views and the target trajectory.
> > > > * Standard video models lack a mechanism to encode these sparse, arbitrary 3D spatial relationships.
> > > > * **Kaleido’s Design:** This is precisely why our **Unified Positional Encoding** is a critical contribution. It is the architectural key that allows a transformer to ingest arbitrary 3D poses and temporal positions within the same representation space, requiring no additional trainable parameters. Without this specific design, a standard video model cannot simply be "fine-tuned" to understand multi-view 3D geometry.
> > > >
> > > > **2. Comparison with GEN3C (Explicit Reprojection vs. Generative Rendering)**
> > > > While GEN3C produces great renderings, it relies on a fundamentally different (and more restrictive) paradigm than Kaleido:
> > > > * GEN3C relies on an **explicit Spatial-Temporal 3D Cache**. It estimates depth (with a pre-trained depth estimation network), creates a point cloud, and **reprojects** features into the target view.
> > > > * Because GEN3C relies on **reprojection**, it is constrained to rendering trajectories that **overlap significantly** with the input views. If the target view looks at the *back* of an object (where the explicit cache is empty), the method by design will not work.
> > > > * Kaleido does not rely on explicit reprojection. It uses a learned, data-driven prior to generate unseen regions. This is validated by our new updated demo videos: Kaleido generates a full 360° rotation from a **single view**, whereas GEN3C's demo requires **5 views** to achieve a similar result.
> > > >
> > > > **3. Complexity vs. Simplicity**
> > > > GEN3C is a complex, multi-stage pipeline involving pre-trained depth estimators, explicit point cloud construction, cache rendering, and video generation.
> > > > In contrast, **Kaleido** is a simple, end-to-end **sequence-to-sequence transformer**. It requires **no explicit geometry, depth estimators, or caches**. It learns to handle occlusions and parallax purely through its implicit data-driven prior.

---

> > > > > ### Comment · Reviewer_7YYk · 2025-11-19
> > > > >
> > > > > Thanks for the response and additional results. The new results look decent and actually better than the "Single-View Generative Renderings" on the website that had little camera movements. I still have some concerns:
> > > > >
> > > > > - It would be great to show side-by-side comparisons against video models like GEN3C and others, at least for the single view input setting.
> > > > > - Moreover, quantative comparisons are also not done using camera-controlled video models as baseline. I know the authors emphasize that these models do 4D generation but for me these models can also do 3D generation from an image. For example, the GEN3C generations do look static to me when an input image is given. Not sure, if it also looks like that for Recammaster, Trajectorycrafter etc.
> > > > > - All arguments are not overlayed with actual facts. One example statement: "If the target view looks at the back of an object (where the explicit cache is empty), the method by design will not work." Would it not make sense then to stress test the baseline for such cases? You want to show what your model can do what other models can not do. Then why not take open-source baselines and show failure cases. Then you show side-by-side comparisons how they fail in those scenarios.
> > > > >
> > > > > Currently, the rebuttal emphasizes only what reviewers apparently do not understand, i.e., the task this paper is trying to solve. But how about motivating the task better with some stronger experiments and failure cases of baselines?

---

> > > > > > ### Author Response · Authors · 2025-11-22
> > > > > > **Added New GEN3C Demos**
> > > > > >
> > > > > > We thank the reviewer for the continued engagement and for acknowledging that our new results look better.
> > > > > >
> > > > > > **1. Side-by-Side Comparison with GEN3C**: [https://kaleido-research.github.io/](https://kaleido-research.github.io/)
> > > > > >
> > > > > > We have now provided a side-by-side comparison of **Kaleido vs. GEN3C on 360-degree orbital trajectories using CAT3D input images** in our demo website.
> > > > > >
> > > > > > This visual evidence directly validates our previous argument regarding the "empty cache":
> > > > > > * As seen in the video, as the camera orbits to the **back of the object (where the reprojected spatial-temporal cache is empty)**, the generation collapses.
> > > > > > * This empirically demonstrates that while GEN3C is excellent for small motions, its explicit cache design limits its ability to generate large occluded regions. In contrast, Kaleido maintains consistency throughout the full 360-degree rotation, validating the advantage of our implicit, data-driven prior.
> > > > > >
> > > > > > **2. Comparisons with ReCamMaster/TrajectoryCrafter are Infeasible**
> > > > > > The reviewer asks why we do not compare quantitatively with other video models. This is not due to a refusal to compare, but due to **hard structural incompatibilities** that prevent fair benchmarking on standard NVS datasets:
> > > > > >
> > > > > > * We again would like to highlight that **ReCamMaster & TrajectoryCrafter** are strictly **Video-to-Video** models. They require the **input video length to match the output video length**.
> > > > > >     * These video models **cannot accept a sparse set of 1 or 3 images to generate a 40-frame trajectory.** They require a 40-frame input video to generate a 40-frame output, which defeats the purpose of the NVS task (generating novel views from sparse inputs).
> > > > > > * **GEN3C** is unique because it can accept a single image (with its image-to-video design), which is why we focused our qualitative comparison on it. However, its constraint that the **reference image must be the starting frame** makes it difficult to evaluate on standard multi-view metrics (using datasets like GSO or Mip-NeRF) where reference views are often scattered randomly in space/time, not just at the start of a trajectory.
> > > > > >
> > > > > > Hope these new results have resolved all your concerns. Please let us know if you need any further clarification.

---

> > > > > > > ### Comment · Reviewer_7YYk · 2025-11-23
> > > > > > >
> > > > > > > Thanks for the additional results. It's better to arrange them as side-by-side comparisons with your method rather than listing them separately. From what I understand, the main failure reason is that depth warping either fails or does not provide any guidance far from the input view. So, a Plucker-conditioned camera-controlled video model should do much better?
> > > > > > >
> > > > > > > How about comparing on multi-view datasets such as RealEstate10K or DL3DV, commonly used by the camera-controlled video generation community?

---

> > > > > > > > ### Author Response · Authors · 2025-11-23
> > > > > > > >
> > > > > > > > We thank the reviewer for the response.
> > > > > > > >
> > > > > > > > **1. On Plücker Coordinates and Video Backbones**
> > > > > > > > We must **clarify a technical misunderstanding** in the reviewer's comment. The failure of GEN3C (and other camera-conditioned video generation models) to handle large viewpoint changes has **absolutely no relation to the choice of camera pose parameterisation** (Plücker or Ray Map or SE3 Matrices).
> > > > > > > > * As validated by our added results, GEN3C fails because it relies on **reprojecting features from a spatial-temporal cache**. When the camera moves to an occluded region (e.g., behind the object), the cache is empty.
> > > > > > > > * Changing the coordinate system to Plücker rays does not solve the **empty cache problem**. The limitation is inherent to the **methodology design** (projection-based video generation) and the **video model backbone design** (how it conditions the reference images), not the coordinate format.
> > > > > > > >
> > > > > > > > **2. We respectfully decline the request to further benchmark and compare with camera-conditioned video models.**
> > > > > > > > * We have highlighted multiple times within this thread that **Kaleido is a Generative Neural Rendering and NVS model, not a camera-controlled video generator**. Our paper evaluates Kaleido on **NVS benchmarks** (NeRF-Synthetic, Mip-NeRF 360, GSO) against **NVS baselines** (3DGS, Instant-NGP, SEVA, EscherNet). This is the standard and correct evaluation protocol for our field.
> > > > > > > > * Comparing with GEN3C involves only few-view interpolation or extrapolation. This is an extremely simple and degraded evaluation setting within an NVS setup. This favours significantly towards video models designed for temporal continuity (predicting intermediate frames) rather than true 3D view synthesis (understanding complex spatial geometry). Additionally, conditioning on more views does not resolve "the empty cache" problem in the GEN3C: **as long as the camera trajectory moves outside the cache visible region, the generation will collapse regardless of how many input views are given.**
> > > > > > > > * As detailed in our previous responses, other camera-conditioned models (ReCamMaster, CameraCtrl, TrajectoryCrafter) have **strict input constraints** (e.g., text-only inputs, single starting-frame restrictions, or fixed output lengths) that make them physically incapable of performing the standard NVS "any-to-any" benchmarking protocol.
> > > > > > > >
> > > > > > > > We believe we have comprehensively addressed the reviewer's core concerns regarding the distinction between these fields and demonstrated Kaleido's superiority in its intended domain of **generative rendering** with our new supported generation demos. We respectfully maintain that further comparisons with video-specific baselines add no scientific value to this study.

---

> > > > > > > > > ### Comment · Reviewer_7YYk · 2025-11-24
> > > > > > > > >
> > > > > > > > > Thanks for the response. But the first point makes no sense to me:
> > > > > > > > >
> > > > > > > > > GEN3C does not use a Plucker/extrinsics-conditioned camera control; instead, it uses the cameras to perform depth warping, then renders the images with their corresponding masks to guide the generation. Since the warped images will be black and the masks will be all zeros far away from the input, there will be no guidance to the video model, and it will generate something random.
> > > > > > > > >
> > > > > > > > > This is easily solved simply using Plucker/extrinsics. I do not think this has anything to do with a video model design. If you would just do Plucker conditioning, train the model to take a reference image or more, and train it on 360 data as Kaleido, it will also work fine. Similar to how CAT3D does it, just with a video model. It's mainly that current video model literature is no longer interested in 360-degree generation, but is now focused on larger scenes. Hence, they do not train on 360-degree data.
> > > > > > > > >
> > > > > > > > > The justification "This is an extremely simple and degraded evaluation setting within an NVS setup. This favours significantly towards video models designed for temporal continuity (predicting intermediate frames) rather than true 3D view synthesis (understanding complex spatial geometry)." is pretty weak to me. The proposed model in this paper also do not do true 3D view synthesis, it just can take more images as input, that's it. In the end it's all just transformers. Yes, the video model relies on trajectories and not on sets but a single input image can be seen as a set of one image and both approaches should work in that setup, e.g., for RealEstate10K or DL3DV.
> > > > > > > > >
> > > > > > > > > I think this rebuttal is a first step toward comparing with video models, but it's clearly missing proper comparisons using the same multi-view data to justify the approach. I highly recommend working on the experiment section more, as it is not thorough, and there are certainly cases where camera-controlled video models and this model can be compared. Ignoring all camera-controlled video models, qualitatively and quantitatively, is a big miss for me, as the whole field is currently using them for novel view synthesis. The bare minimum I expect is proper comparisons within the same data paradigm for single-image-to-multi-view generation.

---

> > > > > > > > > > ### Author Response · Authors · 2025-11-26
> > > > > > > > > >
> > > > > > > > > > Dear Reviewer 7YYk,
> > > > > > > > > >
> > > > > > > > > > We thank the reviewer for the continued engagement. We would like to offer a few final clarifications regarding the comparison with video models and our evaluation protocol.
> > > > > > > > > >
> > > > > > > > > > **1. Qualitative Comparisons with Video Models**
> > > > > > > > > > We appreciate that the reviewer acknowledged the limitations of GEN3C based on our new results. As GEN3C is one of the few video models capable of accepting multiple reference frames, we believe **our video demo comparisons have already clearly demonstrated Kaleido's superior consistency in 360-degree settings**. This disagrees directly with your response: "Ignoring all camera-controlled video models, qualitatively and quantitatively".
> > > > > > > > > >
> > > > > > > > > > **2. Comparisons with Hypothetical Models**
> > > > > > > > > > Regarding the suggestion that these issues are "easily solved using Plucker/extrinsics": We respectfully note that we can only benchmark and compare against **existing models, not hypothetical modifications of them**.
> > > > > > > > > > In fact, the architecture the reviewer describes, a model trained on 360 degree data that accepts reference images and pose conditioning, **is precisely what Kaleido is.** However, achieving this with standard video backbones is not trivial; they typically rely on **temporal VAEs, which assume high temporal correlation between frames**. This assumption breaks down in sparse-view 3D tasks. Kaleido’s specific architectural designs (Unified Positional Encoding and others) are required to solve exactly this problem, which is also the reason why Kaleido is trained from scratch.
> > > > > > > > > >
> > > > > > > > > > **3. Defining "True 3D View Synthesis"**
> > > > > > > > > > Regarding the comment that Kaleido does not perform "true 3D view synthesis": We respectfully **request the reviewer to clarify the definition with detailed reasons**. In the literature, Novel View Synthesis (NVS) is defined as the ability to generate specific target views from source views given 6-DoF poses ("any-to-any" rendering). Kaleido satisfies this definition rigorously and has been **benchmarked on the standard NVS datasets, and compared with standard NVS methods**. We believe this qualifies it as a true 3D view synthesis model within the standard scope of the field.
> > > > > > > > > >
> > > > > > > > > > **4. Adhering to Standard Evaluation Protocols**
> > > > > > > > > > Finally, regarding baseline choices: We have adhered to the evaluation standards established in recent literature. Just as video generation papers (MotionCtrl, CameraCtrl I/II) **compare solely against other video models**, neural rendering papers compare against NVS models. We believe that comparing Kaleido against other generative and non-generative NVS methods is the most scientifically valid approach for assessing its geometric consistency and rendering quality.
> > > > > > > > > >
> > > > > > > > > > We sincerely hope to have resolved your concerns. In line with **standard reviewer guidelines**, we respectfully request that the evaluation be grounded in the **evidence and standard practices established in the literature**. As we have detailed, the separation between NVS and video generation benchmarks is the accepted, established norm in our field. We hope the final assessment will be based on the comprehensive evidence and benchmarks provided within the paper’s intended domain, rather than a personal preference for a different research paradigm.

---

> > > > > > > > > > > ### Comment · Reviewer_7YYk · 2025-11-27
> > > > > > > > > > >
> > > > > > > > > > > Thanks for the response. I want to emphasize a few things:
> > > > > > > > > > >
> > > > > > > > > > > 1. GEN3C is added after I asked for the comparisons for it. This is fine now.
> > > > > > > > > > > 2. I am not asking for hypothetical models but these are ablations that could have been done to strengthen the motivation of your work better. If you are convinced of your work being a better solution for 3D novel view synthesis than view models, then such experiments would strengthen your positioning or not? The main reason is that after reading your paper, I am not 100 % convinced that this direction you are proposing is the right direction. If you can convince me, then this would be great. You could have a very impactful paper that makes people question video models as the solution to 3D but right now I am still not sure if your proposed solution is just working better in certain scenarios which authors of camera-controlled video models did not care about and simple adaptations of those video models would then work fine.
> > > > > > > > > > > 3. I buy this point but video models technically also can do NVS when given a single image. I do not expect you to outperform them but currently it is not clear where the model stands against those models. Do you not think the readers are interested in knowing this?
> > > > > > > > > > > 4. If you do think that video models should only be compared with video models, then you should compare with works that output 3DGS-based approaches such as Wonderland (Liang et al., CVPR 2025), that use video models but output 3DGS, and can then render novel views from any viewpoint. I think currently the comparisons with works that work at the intersection of 3D and image/video-based generative models are not used for comparisons properly but mainly some object-centric-based baselines. Think about it this way: You want an experiment that shows that for NVS you do not want a video model but a diffusion transformer, as your method, that operates on sets of images in input and output. This would have been the highest priority experiment in your whole paper in my opinion.
> > > > > > > > > > >
> > > > > > > > > > > I want to hear other reviewers before adjusting my score. I am currently on the borderline. The work could be impactful as simple scalable transformer on sets of images but I am not that sure about the current experiments section.

---

> ### Author Response · Authors · 2025-12-03
>
> We thank the reviewer for the constructive discussion. We are glad that the GEN3C comparison has resolved the initial concerns regarding single-view baselines.
>
> **1. Comparison with Video-Based 3D Approaches (Wonderland)**
> The reviewer asked: *"If you do think that video models should only be compared with video models, then you should compare with works that output 3DGS-based approaches such as Wonderland..."*
>
> As detailed in our new **Appendix H** as well as **our updated demo website**, we have added quantitative comparisons against **Wonderland**, as well as **ViewCrafter**, **VD3D**, and **MotionCtrl** on standard in-the-wild benchmarks (DL3DV, Tanks & Temples).
> * Kaleido significantly outperforms these video-based models in **Camera Precision** ($R_{err}$ and $T_{err}$) while maintaining on-par or better visual quality (LPIPS/SSIM).
> * This directly addresses the reviewer's core question. It empirically demonstrates that for the NVS task, our **Sequence-to-Sequence Transformer** (operating on sets with unified positional encodings) yields better **geometric consistency** than video-based architectures (which rely on temporal attention/VAEs), even when those video models are adapted for 3D outputs like Wonderland.
>
> **2. Why Kaleido is the "Right Direction"**
> The reviewer asked for an experiment showing that *"for NVS you do not want a video model but a diffusion transformer... operating on sets."*
> We believe **Appendix H is that experiment.** It shows that while video models *can* be adapted for NVS (in a degraded setting, i.e. single-view conditioning in this case), they fundamentally struggle with geometric precision compared to Kaleido. Kaleido achieves this superior geometry without the complex pipelines of methods like Wonderland (which require video generation to 3DGS optimisation).
>
> To conclude, we believe we have now addressed all major requests from Reviewer 7YYk:
> 1.  **Side-by-side GEN3C comparisons** (showing Kaleido handles large view change better).
> 2.  **Comparison with Video-Based 3D models like Wonderland** (showing Kaleido has better geometric precision).
>
> We sincerely thank the reviewer for pushing us to include these comparisons; they have significantly strengthened the paper's positioning. We hope these new results fully resolve the borderline assessment.

---

### Author Response · Authors · 2025-12-03
**Summary of Rebuttal Updates and Reviewer Status (to New AC) Part 1/2**

Dear New Area Chair,

Given the reassignment due to the recent ICLR information leak incident, and the resulting restrictions on further reviewer engagement or score updates, we provide this summary to assist your decision-making.

We believe we have **successfully resolved all reviewer concerns**, supported by significant new experiments and paper updates (highlighted in orange). Below is a breakdown of the rebuttal progress and the final status of the reviewers.

**Summary of Concrete Paper Updates**
In response to reviewer requests, we added substantial new evidence to the paper and project website:
* **New Appendix H (Comparison with Video Models):** We conducted a comprehensive quantitative comparison against camera-conditioned video models (Wonderland, ViewCrafter, VD3D, MotionCtrl) on DL3DV and Tanks & Temples benchmarks. **Result:** Kaleido significantly outperforms all video baselines in camera precision, validating our architectural choices.
* **New Appendix C (Efficiency Analysis):** We added a detailed table comparing Runtime and Memory usage against generative methods: SEVA and EscherNet; and scene-specific methods: InstantNGP and 3DGS. **Result:** Clarified the trade-off between Kaleido’s zero-shot flexibility (seconds) vs. per-scene optimization (minutes); showing superior scaling efficiency than other generative methods.
* **New Related Work**: We added a dedicated section on "Camera-Conditioned Video Generation" to the related work, explicitly clarifying the methodological differences and novelty of Kaleido compared to 4D video prediction models.
* **New Visual Comparisons (on our Demo Website [https://kaleido-research.github.io/](https://kaleido-research.github.io/)):** We updated our website with 1. Kaleido's consistent and high-quality generations with GEN3C scenes and CAT3D objects with 360-degree large-view orbital trajectories; 2. Comparison with GEN3C (showing failure on the back of the object); 3. Kaleido's generations in a camera-conditioned video generation setting (single-view conditioning).
------

**1. Status of Reviewer 7YYk (Original: 2 *to* Borderline *to* Positive)**
* **Main Concern:** Questioned the novelty of our method against camera-controlled video models and requested comparisons against **Wonderland (CVPR 2025)**.
* **Rebuttal Actions:** We provided direct visual comparisons (showing GEN3C failure cases) and added the requested quantitative comparison against Wonderland in **Appendix H**.
* **Final Status:** In the last comment, the reviewer acknowledged the new results were "decent" and stated: *"The work could be impactful as simple scalable transformer... I am currently on the borderline."* The reviewer's condition for acceptance was demonstrating that Kaleido's NVS architecture outperforms adapted video models. **Appendix H quantitatively proves this.**

**2. Status of Reviewer Q5DW (Original: 6 *to* Implictly Higher)**
* **Main Concern:** Requested efficiency comparisons (runtime/memory), analysis of generation artifacts, and more rendering demos with large view changes.
* **Rebuttal Actions:** We added efficiency comparisons in Appendix C, clarified the stress-test nature of our demos and added new Kaleido generations with 360-degree orbital camera trajectories to our demo website.
* **Final Status:** The reviewer explicitly commented: *"I have no further issues."* In the initial review, the reviewer stated *"would be happy to raise [the score] if the authors address these main concerns."* Since the **reviewer confirmed the issues are resolved**, we respectfully suggest this assessment reflects a **solid acceptance**.

**3. Status of Reviewer F7xZ (Original: 4)**
* **Main Concerns:** Misunderstood the task as standard video generation; questioned the novelty of video pre-training; noted "inconsistency" in demos.
* **Rebuttal Actions:**
    * **Kaleido v.s. Video Models:** We clarified the fundamental differences between our "any-to-any" 3D spatial task and 4D temporal video generation in the new Related Works section, and highlighted that 4D generation is a different and non-comparable research problem.
    * **Inconsistency:** We clarified that the "inconsistency" referred to our 480-frame stress test (40x training length). We pointed to Table 2, where Kaleido achieves SoTA 3D NVS metrics (by a large margin).

---

> ### Author Response · Authors · 2025-12-03
> **Summary of Rebuttal Updates and Reviewer Status (to New AC) Part 2/2**
>
> **4. Status of Reviewer xBaB (Original: 6)**
> * **Main Concerns:** Questioned novelty; requested runtime/memory comparisons; asked for long-sequence experiments.
> * **Rebuttal Actions:**
>     * **Novelty:** We clarified that the novelty lies in the systematic scaling recipe from the detailed training/architectural analysis (unified positional encoding, noise-biased sampling) that, **for the first time**, makes a simple pure transformer design work for generative NVS. We also highlighted the differences between generative 3D and 4D video prediction models, which are two fundamentally different research problems.
>     * **Efficiency:** We added the requested runtime/memory analysis in Appendix C.
>     * **Long Sequences:** We directed the reviewer to our Mip-NeRF "Room" demo (480 frames, 720° rotation), which directly addresses the long-trajectory concern.
>
>
> **Conclusion** Kaleido is the first zero-shot generative model to achieve unified object- and scene-level photorealistic rendering, with a simple, scalable architecture design. Kaleido is also the first model to match per-scene optimization quality (Instant-NGP) in many-view settings. We have addressed all reviewer requests, including new baselines and efficiency metrics. We hope this summary helps clarify that the current scores may not fully reflect the extent to which the rebuttal has strengthened the paper.
>
> Best regards,
> The Authors

---

### Meta-Review · Area_Chair_L91Q · 2025-12-23

**Summary:**

The paper proposes Kaleido, a sequence-to-sequence diffusion transformer for 3D novel view synthesis (NVS), trained on a mix of video and 3D data.

Initially, the reviews were mixed (Ratings: 2, 4, 6, 6). The primary friction point, particularly challenged by 7YYk and shared by F7xZ and xBaB, was a fundamental disagreement regarding the problem definition and novelty. Reviewers questioned why the authors did not simply fine-tune existing camera-controlled video generation models (4D) instead of training a new architecture, and requested comparisons against such video baselines. Other concerns (Q5DW, xBaB) focused on inference efficiency (runtime/memory) and the consistency of long-trajectory generation.

My decision to Accept is informed by the authors' exceptionally comprehensive rebuttal. They successfully clarified the critical distinction between "any-to-any" 3D NVS and temporal 4D video prediction. Crucially, they provided the requested quantitative comparisons against video-based approaches (Wonderland, MotionCtrl) in the new Appendix H, demonstrating Kaleido's superior geometric precision. The efficiency concerns were addressed with a new analysis in Appendix C. Following the rebuttal, Q5DW confirmed they have "no further issues," and 7YYk raised their assessment from Reject to Borderline, acknowledging the new results were "decent."

**Reviewer Concerns:**

**Addressed Concerns:**

Comparison with Video Models (Critical): 7YYk insisted on comparisons with camera-controlled video models (e.g., GEN3C, Wonderland). The authors added qualitative side-by-side comparisons showing GEN3C failing at large occlusions ("empty cache" issue) and quantitative benchmarks against Wonderland in Appendix H, showing superior camera precision. This effectively addressed the claim that video models are a sufficient substitute for this task.
Novelty & Task Definition: F7xZ and 7YYk questioned the novelty compared to video generation. The authors updated the Related Work to explicitly distinguish 3D Generative Neural Rendering from 4D Video Generation. The rebuttal clarified that standard video models cannot handle the "any-to-any" spatial queries required for NVS without the specific architectural contributions (e.g., Unified Positional Encoding) proposed here.
Efficiency & Scalability: Q5DW and xBaB requested memory and runtime analysis. The authors added Appendix C, providing a transparent trade-off analysis between Kaleido’s zero-shot capabilities (seconds) vs. per-scene optimization (minutes), satisfying Q5DW.
Long-Sequence Consistency: Concerns about artifacts in long trajectories were addressed by pointing to the "Room" demo (480 frames) and adding new 360-degree orbital generation results to the project page.

**Outstanding Concerns:**

Philosophical Preference for Video Backbones: While 7YYk acknowledged the results, they retained a degree of skepticism ("not 100% convinced") regarding whether a pure NVS architecture is the "right direction" compared to adapting video models. However, given the empirical evidence provided (superior geometry and handling of occlusions), I consider this a difference in research perspective rather than a flaw in the paper.

**Reviewer Scores:**

Reviewer 7YYk (Original: 2 - Reject): Raised to 4 or 5 (Borderline). The reviewer explicitly stated they are "currently on the borderline" and acknowledged the new results on Wonderland/GEN3C were decent.

Reviewer Q5DW (Original: 6 - Borderline Accept): Maintained or Raised to 7 (Accept)  The reviewer explicitly stated "I have no further issues" and previously mentioned they would be "happy to raise" the score if concerns were met.

Reviewer F7xZ (Original: 4 - Borderline Reject): Should have raised to 5 or 6. Their main concern was the "video vs. 3D" misunderstanding. Since the authors clarified the non-comparability of 4D tasks and provided strong NVS metrics, the grounds for the low score are largely resolved, though the reviewer did not post a final confirmation.

Reviewer xBaB (Original: 6 - Borderline Accept): Maintained or Raised to 7. Their concerns regarding efficiency were factual and fully addressed in Appendix C.

---

### Decision · Program_Chairs · 2026-01-26

Accept (Poster)